# In-organoid single-cell CRISPR screening reveals determinants of hepatocyte differentiation and maturation

Junbo Liang[1†], Jinsong Wei[2†], Jun Cao[1,3†], Jun Qian[1], Ran Gao[1], Xiaoyu Li[2], Dingding Wang[1], Yani Gu[1], Lei Dong[4], Jia Yu[1], Bing Zhao[5,6,7*] and Xiaoyue Wang[1,3*]

†Junbo Liang, Jinsong Wei and Jun Cao contributed equally to the work.

*Correspondence:
bingzhao@biogenous.cn;
wxy@ibms.pumc.edu.cn

[1] State Key Laboratory of Common Mechanism Research for Major Diseases, Department of Biochemistry and Molecular Biology, Institute of Basic Medical Sciences Chinese Academy of Medical Sciences, School of Basic Medicine Peking, Union Medical College, Beijing 100005, China
[5] School of Basic Medical Sciences, Jiangxi Medical College, Nanchang University, Nanchang 330031, China
Full list of author information is available at the end of the article

## Abstract

**Background:** Harnessing hepatocytes for basic research and regenerative medicine demands a complete understanding of the genetic determinants underlying hepatocyte differentiation and maturation. Single-cell CRISPR screens in organoids could link genetic perturbations with parallel transcriptomic readout in single cells, providing a powerful method to delineate roles of cell fate regulators. However, a big challenge for identifying key regulators during data analysis is the low expression levels of transcription factors (TFs), which are difficult to accurately estimate due to noise and dropouts in single-cell sequencing. Also, it is often the changes in TF activities in the transcriptional cascade rather than the expression levels of TFs that are relevant to the cell fate transition.

**Results:** Here, we develop Organoid-based Single-cell CRISPR screening Analyzed with Regulons (OSCAR), a framework using regulon activities as readouts to dissect gene knockout effects in organoids. In adult-stem-cell-derived liver organoids, we map transcriptomes in 80,576 cells upon 246 perturbations associated with transcriptional regulation of hepatocyte formation. Using OSCAR, we identify known and novel positive and negative regulators, among which *Fos* and *Ubr5* are the top-ranked ones. Further single-gene loss-of-function assays demonstrate that *Fos* depletion in mouse and human liver organoids promote hepatocyte differentiation by specific upregulation of liver metabolic genes and pathways, and conditional knockout of *Ubr5* in mouse liver delays hepatocyte maturation.

**Conclusions:** Altogether, we provide a framework to explore lineage specifiers in a rapid and systematic manner, and identify hepatocyte determinators with potential clinical applications.

**Keywords:** Hepatocyte differentiation and maturation, Organoid, Single-cell CRISPR screen, Regulon

## Background

The liver is the largest internal organ that controls metabolism, secretion, and detoxification, and those essential functions are mainly executed by hepatocytes [1]. Deciphering the mechanism of hepatocyte differentiation and maturation facilitates regenerative medicine, which is aimed to promote the generation of functional hepatocytes in vitro or in vivo for the treatment of liver diseases. Functional hepatocytes are generated through tightly controlled transcriptional programs in vivo. During liver development, the liver progenitor hepatoblasts are specified from the foregut of endoderm and then differentiate into hepatocytes and cholangiocytes [2]. Hepatocytes derived from liver progenitor-like cells are also observed in vivo, when the hepatic parenchyma is severely compromised during injuries [3–5]. It is known that transcription factors (TFs) such as *FOXA2*, *HNF4α*, and *CEBPA* are required for the hepatocyte differentiation and maintenance of functional mature hepatocytes, and loss of any of those key TFs will lead to the degeneration of hepatocytes and liver diseases [2]. Based on the current understanding of hepatocyte specification, step-wised differentiation of hepatocytes from human pluripotent stem cells (PSC) has been established [6, 7]. However, the PSC-derived hepatocytes are not functionally mature, implying the unknown realm of hepatocyte differentiation and maturation.

Single-cell profiling of fetal liver development has revealed thousands of genes, including dozens of TFs that are dynamically expressed during hepatocyte formation [8, 9]. To identify key regulators for hepatocyte differentiation and maturation from a large number of candidates, a high-throughput assay in a near-physiological model is urgently needed. Compared to 2D culture models, adult-stem-cell-derived 3D organoids possess advantages in experimental manipulability and scalability, as well as capturing aspects of the native tissue architecture and function in vitro [10]. Organoids derived from intrahepatic cholangiocyte organoids (ICOs) [11] are facultative and could convert into hepatocytes in defined culture conditions [12–14]. This ICO-based one-step hepatocyte differentiation system has been successfully applied to test a small pool of hepatocyte specifiers through a gain-of-function assay in our previous study [15]. However, only the expression of Albumin was used as the readout, and the heterogeneity of organoids was not accounted for, limiting the systematic discovery of hepatocyte regulators that are involved in the complex process.

Single-cell CRISPR (scCRISPR) screens, in which single-cell transcriptomes are used as readouts for CRISPR gene perturbations, are suitable to assess transcriptional effects on cell fate or cell states for vast candidate genes [16–22]. scCRISPR screens have not been applied in 3D organoids for lineage specifiers at large scale. While the rich scRNA-seq readout would capture the heterogeneity of organoids, it also poses additional challenges for determining the perturbation effects that are required for the identification of key transcriptional regulators. Cell fate transition is often driven by a cascade of transcriptional activation while each TF may only present at low abundance at a given time point. Due to the noise and dropouts in single-cell sequencing, it is difficult to accurately estimate the change of low abundance transcripts. In addition, sometimes it is the change of TF activities, rather than a change in TF abundance, that initiates transcriptional cascades. Most existing analysis tools for scCRISPR screens estimate the direct perturbation effects on gene expression [16, 18, 23], rather than on TF activities. It has

been shown that using changes in the activities of regulons (the TF-centered gene regulatory modules) is a more robust method for cell state identification than using differential gene expression in scRNA-seq [24]. Therefore, using the changes in the activities of regulons to assess the perturbation effects is a possible way to address the challenges for TF regulator screens in organoids.

Here, we established OSCAR, a method for scCRISPR screening in organoids using changes in regulatory network as readouts and applied it to identify modulators of hepatocyte differentiation and maturation. Using OSCAR, two sets of modulators that potentiated or attenuated this process were identified and ranked on their regulatory effects on hepatocyte marker expression. Among the top modulators, c-Fos, a component of activated protein-1 (AP-1) transcription factor, caused accelerated hepatocyte differentiation and maturation when perturbed in both mouse ICOs (mICOs) and human ICOs (hICOs); whereas *Ubr5* loss led to an opposite effect. Collectively, our study demonstrated an organoid-based scCRISPR screen approach that was scalable, manipulable, and general purpose for identifying regulators in cell fate transition.

## Results

### Establishment of an organoid-based single-cell CRISPR screen for hepatocyte lineage regulators

To assess the feasibility of conducting single-cell CRISPR screens in 3D organoids, we performed a proof-of-concept CROP-seq screen in mouse ICOs (mICOs), isolated from livers of mice expressing spCas9-EGFP at Rosa26 locus (Fig. 1a) [25]. mICOs form a self-organized cystic structure in the expansion medium (EM) with the expression of hepatocyte lineage markers as well as ductal markers. mICOs could convert into hepatocyte-like cells in a defined differentiation medium (DM) for ~ 12 days [12, 14]. To improve the scalability and manipulability of the system for setting up the in-organoid single-cell CRISPR screen, we shortened the duration of the differentiation strategy from 12 days (DM_12) to 7 days (DM_7) and compared the transcriptional profiles between them. In agreement with previous reports [12], both strategies could induce comparable expression of multiple hepatocyte markers such as *Alb*, *Ttr*, *Cyp3a11*, *Apoa1*, *Mup20*, *Mrp2*, *Sult1a1*, and *Aldh1a1* as well as the reduction of ductal markers such as *Sox9* and *Spp1*, indicative of cholangiocyte-to-hepatocyte transition (Additional file 1: Fig. S1a). Principal component analysis (PCA) of transcriptional profiles revealed that the samples from DM_7 clustered closest to those from DM_12, indicating the overall transcriptional similarity between the two strategies (Additional file 1: Fig. S1b). Differential gene expression analysis revealed that ~ 80% of the upregulated genes and downregulated genes are conserved between DM_7 and DM_12 groups (Additional file 1: Fig. S1c and Additional file 2: Table S1a, b). Nearly, all of the gene ontology (GO) biological processes enriched for upregulated genes shared in DM_7 and DM_12 groups were metabolism-related, while the enriched processes associated with the downregulated genes were related to the cell cycle pathway (Additional file 1: Fig. S1d and Additional file 2: Table S1c, d). Gene set enrichment analysis (GSEA) of transcriptional profiles for the DM_7 strategy identified multiple metabolism-related processes such as retinol metabolism, drug metabolism, and complement and coagulation cascades (Additional file 1: Fig.

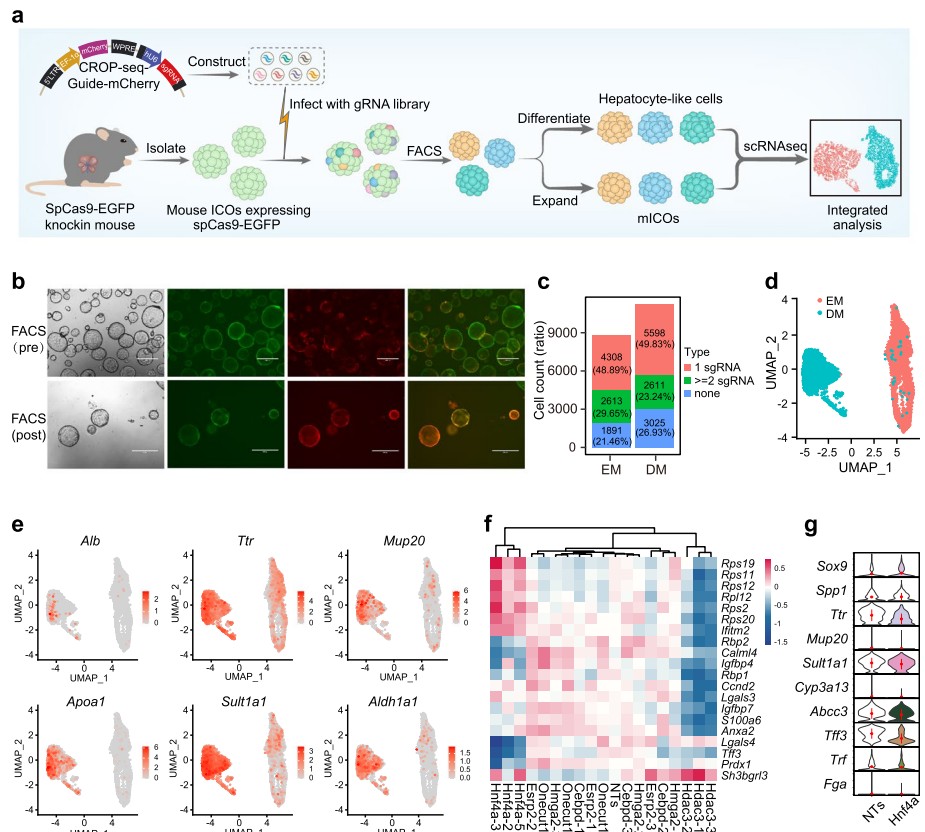

**Fig. 1** Establishment of an organoid-based single-cell CRISPR screen for hepatocyte lineage regulators. **a** Schematic design of the organoid-based CROP-seq screen. **b** Mouse ICOs isolated from Cas9-EGFP knockin mice were transduced with the pilot CROP-seq screen lentiviral library carrying a mCherry reporter (pre-FACS; top panel). At day 3 post-transduction, mICOs were enriched with FACS for genome-edited cells (GFP$^+$/mCherry.$^+$) and expanded (post-FACS; bottom panel). Scale bars, 500 μm. **c** The number and corresponding ratio of cells expressing a unique sgRNA (red), more than two sgRNAs (green) or none (blue) in the EM and DM groups. **d** UMAP visualization of cells with unique sgRNA assignments in the EM and DM groups. **e** UMAP visualization of the expression of known hepatic markers, including *Alb*, *Ttr*, *Mup20*, *Apoa1*, *Sult1a1*, and *Aldh1a1*. Expression is color-scaled in log2 for each cell. **f** Heatmap graph displaying the perturbation effect of each sgRNA on the top 20 variable genes. **g** Violin plots showing the relative expression of known hepatocyte and cholangiocyte markers in *Hnf4a*-KO single cells in the pilot screen. Color filling indicates a *P*-value less than 0.05 (Wilcoxon rank sum test)

S1e-h), suggesting the differentiation of mICOs into functional hepatocyte-like cells. Together, the above results demonstrated that a 7-day differentiation strategy was adequate for producing hepatocyte-like cells.

For the pilot screen, we designed a sgRNA library targeting six hepatic fate regulators (three sgRNAs per gene), namely *Hnf4α*, *Onecut1*, *Hdac3*, *Esrp2*, *Cebpd*, and *Hmga2*, as well as 4 non-targeting controls (NTs), for a total of 22 sgRNAs (Additional file 2: Table S2). The sgRNAs were individually cloned into the lentiviral CROP-seq vector modified with an mCherry reporter. To perform the screen, mICOs were transduced with lentivirus produced from this CROP-Guide-mCherry gRNA library at a multiplicity of infection (MOI) of 0.3, and enriched with FACS for genome-edited cells (GFP$^+$/mCherry$^+$) at day 3 post-transduction (Fig. 1a, b). At day 5 post-sorting, the GFP$^+$/mCherry$^+$ double positive mICOs were split and subjected to either EM for

expansion or DM for differentiation for 7 days; and both cell populations were captured using the 10 × Genomics scRNA-seq 3' polyA-primed platform (Fig. 1a).

After sgRNA assignments and scRNA-seq quality controls based on the number of genes expressed, the count of RNA and the percentage of mitochondrial genes (Additional file 1: Fig. S2a, b), we obtained a total of 8812 cells in the EM group and 11,234 cells in the DM group (Fig. 1c). Over 70% of the cells in each group had at least one sgRNA assignment, and ∼50% had a unique sgRNA assigned, indicating the low multiplicity of infection (MOI) (Fig. 1c and Additional file 1: Fig. S2c). To eliminate the confounding effect, we exclude any cells that were assigned to multiple sgRNAs from downstream analysis. The cells with unique sgRNA assignments were normalized and scaled for clustering and dimensional reduction by uniform manifold approximation and projection (UMAP), visualizing a clear separation of cells from EM and DM groups (Fig. 1d). A cohort of known hepatocyte marker genes, such as *Alb*, *Ttr*, *Mup20*, *Apoa1*, *Sult1a1*, and *Aldh1a1*, were upregulated in cells of DM group (Fig. 1e), confirming response to differentiation induction. When we clustered the sgRNAs based on their effects on the top 20 variable genes computed by scMAGeCK [23], sgRNAs targeting the same gene had similar perturbation effects (Fig. 1f). Interestingly, sgRNAs targeting *Hdac3* resulted in an opposite expression pattern compared to those targeting *Hnf4α*, which is consistent with their opposite roles in hepatocyte differentiation and metabolism [26, 27], albeit classic hepatocyte or cholangiocyte markers have not been detected in those top 20 variable genes (Fig. 1f). Violin plots of several hepatocyte or cholangiocyte markers further confirmed the inhibitory effect of *Hnf4α*-KO on cholangiocyte-to-hepatocyte differentiation (Fig. 1g). Those results suggested that the single-cell transcriptome profiles in mICOs can reveal the perturbation effects for key regulators of hepatic fate determination.

## OSCAR for identifying novel modulators regulating hepatocyte differentiation and maturation

Next, we applied our screening strategy to an extensive gene set of candidate modulators shortlisted by either comparing the bulk RNA-seq data of mouse liver E14.0 with that of adult mouse liver or mining previously reported regulators involved in liver development [27–32]. This library includes 236 sgRNAs targeting 79 genes (3 sgRNAs per gene except for Hnf4α with 5 sgRNAs) along with 10 NTs, resulting in 246 sgRNAs in total (Additional file 2: Table S3). Similar to the pilot screen, spCas9-EGFP mICOs were transduced with the sgRNA library of 246 sgRNAs at a representation of 2000 cells per sgRNA, as only a fraction of dissociated mICOs generated new organoids [33]. We sequenced the transcriptome of 80,576 cells at a median depth of ∼50,000 reads per cell. After assigning sgRNAs to each individual cell and filtering out low-quality cells, we obtained 30,217 cells in the DM group and 11,166 cells in the EM group with unique sgRNA assignments, accounting for more than 50% in their respective group, which is similar to the percentages in pilot screen (Fig. 2a and Additional file 1: Fig. S3).

To identify regulators in the differentiation and maturation of hepatocytes, we developed a computational analysis pipeline (OSCAR) for assessing the perturbation effects based on changes in regulatory networks (Fig. 2a). As the differentiation process is often a transcription factor (TF)-driven cascade, we reasoned that the activities of TF-centered co-varying gene modules were the most biologically meaningful readout for our

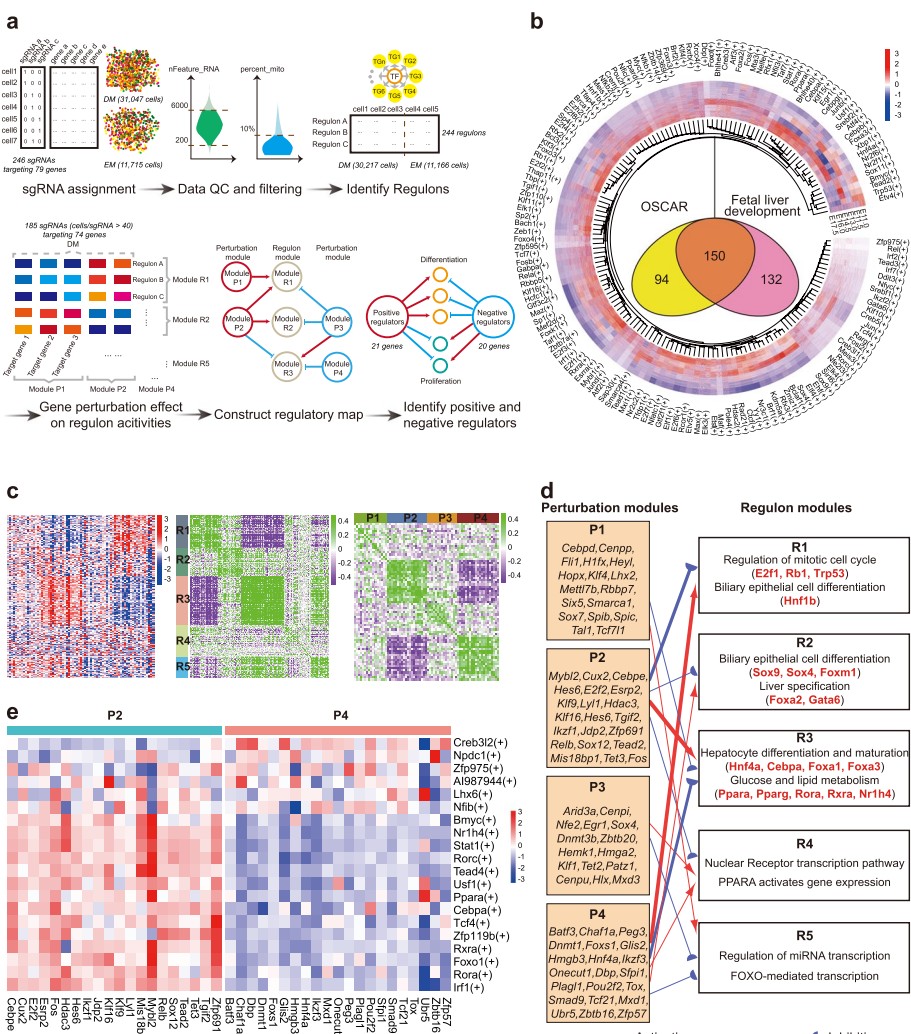

**Fig. 2** CROP-seq screen in mICOs identifies novel modulators regulating hepatocyte differentiation and maturation. **a** Schematic illustration of the OSCAR analysis pipeline. **b** Overlap of the regulons in our OSCAR screen and those differentially activated in mouse fetal livers throughout development [8]. The circos map contains the following layers from outside to inside: the master transcription factor of the 150 overlap regulons, the relative activity score of the indicated regulon at embryo stage E11.0, E11.5, E13.0, E14.5, E16.0, and E17.5, respectively. **c** Representation of perturbation modules and regulon modules. Left heatmap: The effect size on each regulon feature (rows) after perturbation of each candidate gene (columns). Middle and Right heatmap: *k*-means clustering of Spearman's correlation coefficient of regulons (middle, *k* = 5) and candidate genes (right, *k* = 4) computed from the effect size matrix. **d** Regulatory relationships between perturbation modules and regulon modules. The thickness of the line represents the −log10 of the *P*-value of hypergeometric tests. Only lines with the *P*-value less than 0.05 were retained. **e** Heatmap graph showing the perturbation effects of each candidate gene in module P2 and P4 in OSCAR screen on the activities of the top 20 variable regulons

screen. Evaluating the transcriptional changes at the gene-module level instead of the single-gene level also reduces the noises raised from the stochastic nature of single-cell expression data [34]. Therefore, we mapped such TF "regulons" de novo from our screen data by considering both gene co-expression and TF-gene regulatory relationships using SCENIC [35] (Fig. 2a). In total, we have uncovered 244 regulons whose activities varied significantly in our OSCAR screen (Additional file 2: Table S4). Among them, 150

regulons shared the master transcription regulators with the 282 regulons that were differentially activated in mouse fetal livers throughout development (Fig. 2b). The master transcription factor of these regulons includes known key regulators of hepatocyte differentiation, such as *Hnf4α*, *Foxa2*, and *Cebpα*, as well as transcription factors activated during cholangiocyte differentiation, such as *Hnf1β*, *Sox4*, and *Sox9*. In addition, the top 20 variable regulons derived from our data are associated with terms of differentiation and biosynthesis, including multiple lipid and glucose metabolism pathways (Additional file 1: Fig. S4 and Additional file 2: Table S5). These results suggested that perturbations in liver organoids are capable to induce changes in the regulatory networks that recapitulate the in vivo changes during hepatocyte differentiation and maturation.

To evaluate our regulon-based analysis method, we first compared the robustness of regulon activity with gene expression by downsampling the cells in our data. We found that only 40–50 cells are needed to reliably detect changes in regulon activities, while it took 120–150 cells to robustly identify gene expression changes (Additional file 1: Fig. S5). These results suggested that the regulon activity is more robust than single-gene expression due to the sparsity of single-cell sequencing data.

Subsequently, we compared the performance of OSCAR in profiling perturbation effects with two other methods, scMAGeCK and MIMOSCA (See "Methods"). We used the correlation coefficient of the top 200 variable features for each perturbation in relative to *Hnf4a* as the metrics. The three methods are highly consistent with each other, with Spearman's correlation coefficients ranging from 0.89 to 0.97 (Additional file 1: Fig. S6a-c). Next, we measured the robustness of the method, by comparing the agreement of perturbation effects calculated from different subsets of the data with those calculated from the whole dataset. OSCAR and scMAGeCK are comparable in robustness, while MIMOSCA had a higher variation among different rounds of subsampling (Additional file 1: Fig. S6d). From the robustness score, we estimated that the cell number required for obtaining robust perturbation effects is 30 to 40% for OSCAR and scMAGeCK compared to the cell number for the original version of MIMOSCA.

We found that the GO terms associated with the master TFs from the top variable regulons identified in OSCAR are more relevant to liver function than the top variable TFs identified in scMAGeCK or MIMOSCA (Additional file 1: Fig. S6e). Similarly, OSCAR revealed more regulatory changes associated with cell differentiation in the ESC CROP-seq data (Additional file 1: Fig. S6f). These results suggested that regulon-based analysis could better reveal biological insights from changes in the whole regulatory network than the gene expression-based analysis.

To identify the regulators of hepatocyte formation, we clustered the gene perturbations by the mean regulon activity across all the perturbed cells after removing the sgRNAs with less than 40 targeted single cells (Fig. 2a). Using *k*-means clustering, we partitioned the perturbed genes into 4 perturbation modules (P1-P4) and regulons into 5 programs (R1-R5) (Fig. 2c). We also inferred a regulatory network between perturbation modules and regulon programs from the data (Fig. 2d). Interestingly, genes in group P2 and group P4 showed more significant regulatory effects than other groups (Fig. 2d). Perturbations in P4 inhibited the regulon program R3, featuring terms such as hepatocyte differentiation, glucose and lipid metabolism. In contrast, perturbations in P2 activated R3 but inhibited the activities of R1 and R2, which are closely associated

with biliary epithelial cell differentiation and mitotic cell cycle (Fig. 2d). We observed the well-known differentiation enhancer *Hnf4α* and repressor *Hdac3* located in P4 and P2, respectively [26, 27, 36], suggesting that P4 may represent the group of positive regulators of hepatocyte formation and P2 represented the negative ones (Fig. 2d).

We next assessed the association of each perturbed gene with the activities of the top 20 variable regulons in each regulon program (Additional file 1: Fig. S7). We found that perturbations in group P2 increased the activities of Foxa3 and Cebpa-centered regulons, which are known to be key transcription factors involved in hepatocyte specification and differentiation [37], as well as the Ppara and Nr1h4-centered regulons, which are involved in metabolic regulation [38, 39] (Fig. 2e and Additional file 1: Fig. S7). On the contrary, perturbation of the genes in group P4 had the opposite effects on these regulons (Fig. 2e and Additional file 1: Fig. S7). Among those top 20 variable regulons, we also found key TFs of signaling pathways that determine liver cell fate: Tead4 is recognized as a DNA-anchor protein of the YAP transcription complex at the downstream of Hippo-YAP signaling, which controls cholangiocyte specification and hepatocyte trans-differentiation [33, 40–42]. Tcf7l2/Tcf4 acts as the transcriptional partner with β-catenin at the downstream of Wnt signaling, which is also required for liver development and the postnatal proliferation of hepatocytes [2, 33].

Together, these results suggested our computational framework for scCRISPR data analysis is reliable for identifying key cell fate regulators and infer their biological functions from changes in the whole regulatory network.

### Modulators identified from OSCAR could be ranked based on the perturbation effects on hepatocyte marker expression

To further explore the function of the candidate regulators in modules P2 and P4, we sought to map the perturbation effects on hepatocyte differentiation at the single-gene expression level. Due to the variability in editing outcomes of the CRISPR-Cas9 system, the presence of a sgRNA in a cell does not always indicate the knockout of the target gene. To accurately map the perturbation effects, we applied the MIMOSCA framework to infer the perturbation probability for each cell [18] (Fig. 3a). For non-targeting controls, the distribution of the perturbation probability follows a normal distribution with a mean of 0.5. In contrast, bimodal distributions of the perturbation probability were observed for the sgRNAs targeting the candidate regulators, suggesting the separation of unperturbed and perturbed cells (Fig. 3b). To identify those unperturbed cells, we fitted a two-component Gaussian mixture model on the perturbation probability distribution for each gene in the P2 and P4 group (Fig. 3a and Additional file 1: Fig. S8). After removing the unperturbed cells, we found an increased expression pattern of 9 hepatocyte-specific marker genes (including *Alb*, *Ttr*, *Tff3*, and *Cyp3a13*) in the P2 perturbation group (Fig. 3c), and the opposite expression pattern in the P4 perturbation group (Fig. 3d). These results support that perturbation of these candidate genes directly impact the emergence of mature hepatocyte function, such as the secretion of Albumin and expression of cytochrome 450 enzymes. Therefore, we integrated the expression of the 9 hepatocyte markers into a hepatocyte regulator score, by which the candidate modulators in P2 and P4 could be ranked (Fig. 3e, f).

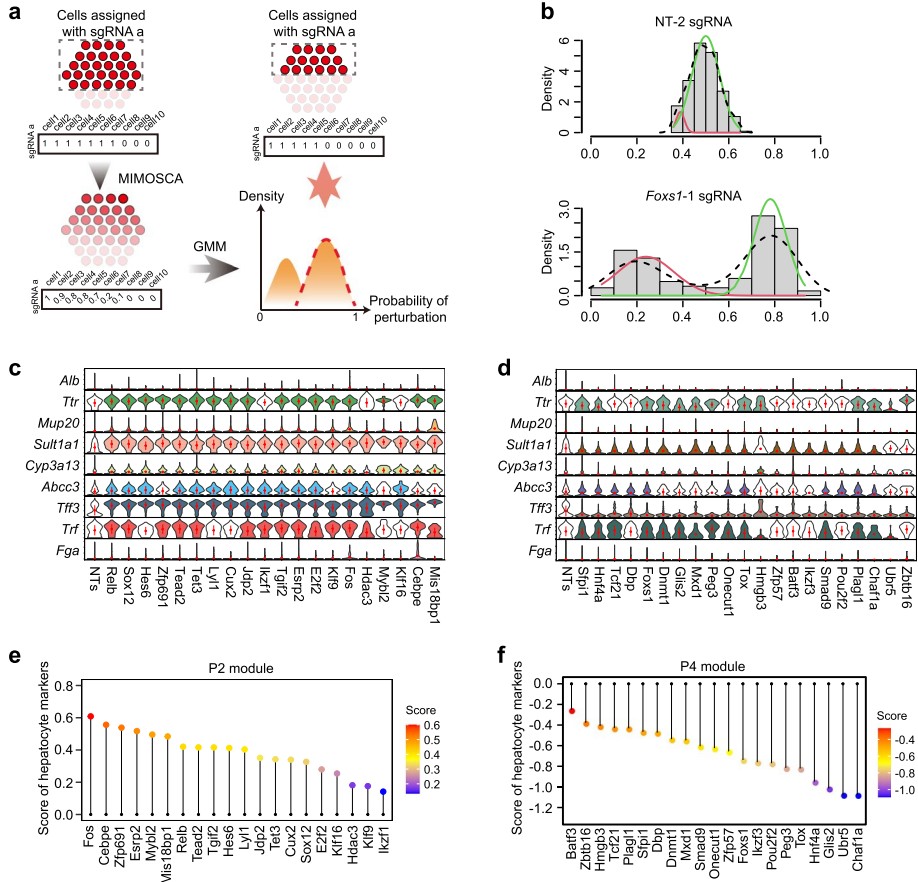

**Fig. 3** Candidate modulators identified from the screen regulate hepatocyte marker expression. **a** Schematic illustration of the strategy to filter unperturbed cells for expression analysis. **b** The representative distribution curve of perturbation probability of cells with sgRNA targeting NT-2 (non-targeting) and *Foxs1*-1 sgRNA. **c**, **d** Violin plot showing the relative effects of each perturbation from module P2 (**c**) or module P4 (**d**) on selected known hepatic markers compared to NT controls. Color filling indicates a *P*-value less than 0.05 (Wilcoxon rank sum test). **e**, **f** Perturbation score of candidate modulators in P2 (**e**) and P4 (**f**) groups using hepatocyte regulator score calculated with expression change of the nine hepatocyte markers used in **c** and **d**

### Deletion of *Fos* boosts hepatocyte differentiation by upregulating metabolic pathways

Among the top candidates, *Fos* perturbation exhibited the highest hepatocyte regulator score (Fig. 3e). To further investigate the role of *Fos* in hepatocyte differentiation and maturation, we generated stable *Fos* KO mICO lines and compared them to the non-targeting (NT) control mICOs under EM or DM conditions (Fig. 4a, b). Morphologically, *Fos* KO organoids closely resembled the NT mICOs (Supplementary Figure Additional file 1: Fig. S9a). Using the cell viability assay, we found that their growth rates were comparable (Additional file 1: Fig. S9b). In *Fos* KO organoids cultured with expansion medium, cholangiocyte markers (*Sox9* and *Spp1*) were upregulated, while hepatocyte markers were barely detectable by qRT-PCR (Additional file 1: Fig. S9c). Upon induction with differentiation medium, globally, cholangiocyte markers were significantly reduced in NT and *Fos* KO organoids, while all hepatocyte markers (*Alb*, *Ttr*, *Apoa1*, *Mup20*, and *Mrp2*) were significantly increased (Additional file 1: Fig. S9c). Interestingly, four of the five hepatocyte markers were expressed at significantly higher levels in the *Fos* KO

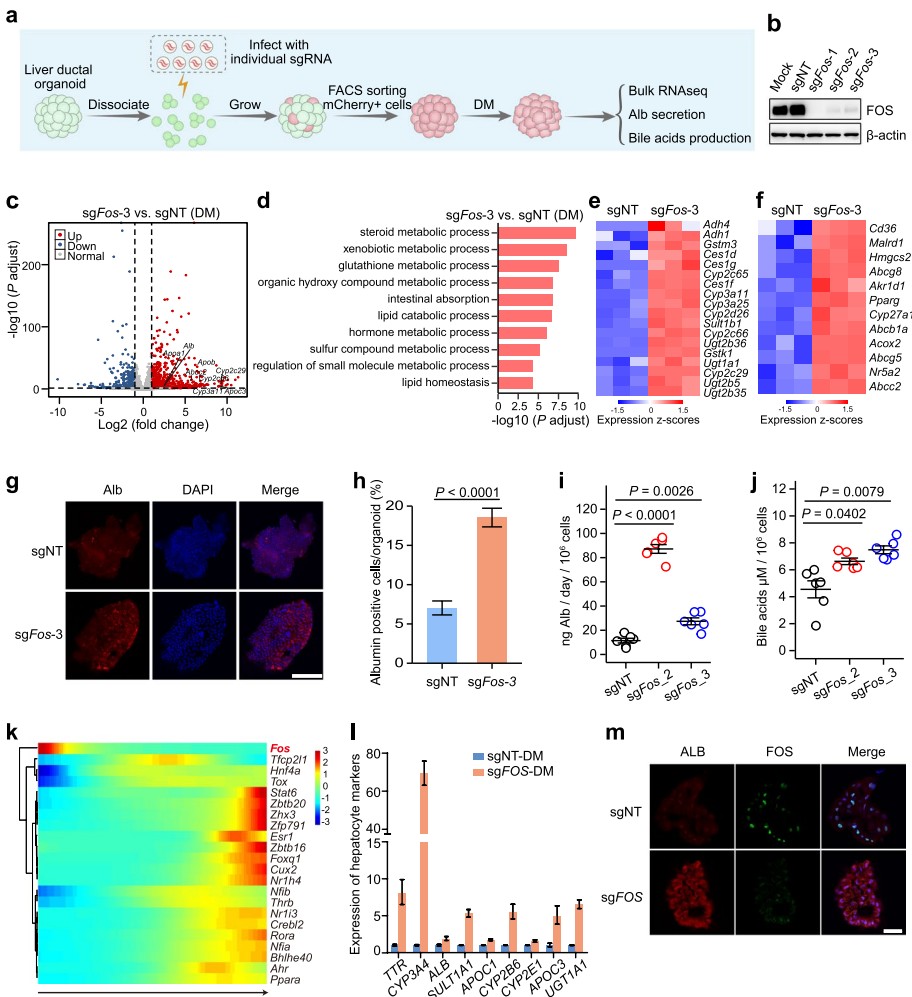

**Fig. 4** *FOS* perturbation boosts hepatocyte differentiation and maturation in ICOs. **a** Flowchart of individual validation of *Fos* perturbation in ICOs. **b** Western blot analysis of Fos protein levels in mICOs transduced with NT sgRNA or sgRNA targeting *Fos*. Uncropped images of the Western blots are available in Additional File 1: Fig. S12. **c** Volcano plot of differentially expressed genes (|log2FoldChange|< 1 and an adjusted *P*-value using the Benjamini-Hochberg (BH) method < 0.05) in *Fos* KO mICOs compared with NT control ICOs in DM condition. **d** The top 10 enriched GO-BP terms (sorted by adjusted *P*-values using the BH method) of significantly upregulated genes in *Fos* KO mICOs compared with the NT controls in DM condition. **e**, **f** Heatmap showing relative expression of genes involved in xenobiotic metabolism (**e**) or lipid metabolism (**f**). **g** Representative immunofluorescent staining of Alb expression in *Fos* KO mICOs compared with NT control ICOs in DM condition. Scale bars, 100 μm. **h** Percentages of Albumin positive cells for mICOs in **g** were shown as mean ± s.e.m. (*n* = 5) and compared by two-tailed Student's *t* test. A *P*-value less than 0.05 was considered statistically significant. **i**, **j** Secretory Alb protein levels (**i**) and bile acid production (**j**) were measured in *Fos* KO mICOs compared with the NT controls in DM condition. Data were shown as mean ± s.e.m. (*n* = 6). Following one-way ANOVA, the Bonferroni method was used as the post hoc test for pairwise comparisons. Statistical significance was defined as a *P*-value below 0.05. **k** The expression dynamics of *Fos*, *Hnf4a*, and 20 reported TFs upregulated during hepatocyte development [43] using the previously published single-cell RNA-seq data [8]. **l** qPCR analysis showing the expression of hepatocyte genes in human *FOS* KO ICOs compared with the NT controls in DM conditions. **m** ALB protein levels as measured by immunofluorescent staining in human *FOS* KO ICOs compared with the NT controls in DM conditions. Scale bars, 50 μm

organoids. These results suggested that *Fos* is a conditional liver cell fate regulator, and the deletion of *Fos* significantly enhanced hepatocyte differentiation upon differentiation induction.

To further understand the molecular mechanisms underlying the observed effects of *Fos* knockout, we performed RNA-seq for *Fos* KO and NT mICOs that underwent differentiation induction. Differential gene expression analysis of RNA-seq data revealed 840 upregulated genes and 397 downregulated genes in *Fos* KO mICOs (Fig. 4c and Additional file 2: Table S6a). As expected, multiple common hepatocyte markers such as *Alb*, *Apoa1*, *Apob*, *Apoc2*, *Cyp2c29*, *Cyp2c66*, and *Cyp3a11* were among the upregulated genes (Fig. 4c). Interestingly, functional analysis showed enrichment of upregulated genes in multiple hepatic function-related processes, including lipid metabolism, xenobiotic metabolism, and glutathione metabolism (Fig. 4d and Additional file 2: Table S6b). Specifically, major metabolic enzymes such as cytochrome P450 (Cyp450) and UDP-glucuronosyltransferase (UGT) accounting for detoxification were significantly enriched (Fig. 4e and Additional file 2: Table S6b). In addition, we observed representative genes involved in fatty acid metabolism (*Cd36*, *Pparg*), bile acid homeostasis (*Cyp27a1*, *Akr1d1*, *Malrd1*, *Nr5a2*), and *Abc* transporters for bile secretion (*Abcc2*, *Abcg5*, and *Abcg8*) were significantly upregulated (Fig. 4f and Additional file 2: Table S6b). In agreement with an increase in *Alb* mRNA level, immunofluorescent staining revealed an elevated protein level in *Fos* KO mICOs compared to that of the NT mICOs (Fig. 4g, h). *Fos* KO mICOs also displayed a significant increase in Alb secretion and bile acid production (Fig. 4i, j), suggesting that they were more mature in terms of hepatocyte functions.

Since *Fos* knockout led to accelerated hepatocyte differentiation in our mouse ICO model, we ask whether *Fos* play a role in hepatocyte differentiation in vivo. By analyzing *Fos* expression pattern during liver development, we found that *Fos* is highly expressed in E11.0, but dramatically decreases afterward (Fig. 4k), which is contrary to the tendency of *Hnf4α* and 20 reported TFs that are upregulated during hepatocyte development [43], suggesting the *Fos* suppression is required for the onset of hepatocyte differentiation [8]. On the other hand, *Fos* deletion had been proved to upregulate metabolic pathways in a liver conditional knockout mouse model [44], which is consistent with our data obtained from the organoid model. Collectively, we proposed that *Fos* is a suppressor for hepatocyte differentiation and maturation.

### *FOS* perturbation in human ICOs potentiates hepatocyte differentiation

To explore whether the augmented effects on hepatocyte differentiation and maturation upon the *Fos* deletion is conserved across mouse and human, we evaluated the effects of human *FOS* depletion in the human intrahepatic cholangiocyte organoid (hICO) model. To this end, we first generated hICOs by isolating bile-duct epithelial cells from donors receiving hepatectomy. hICOs are similar with mICOs in expression profiles and the ability to convert to hepatocyte-like cells in defined culture medium [12–14]. Human ICOs were then transduced with an all-in-one lentivirus expressing both Cas9 and sgRNA targeting *FOS*, to generate *FOS*-deleted organoids, which were subjected to differentiation using a strategy similar to the mICOs differentiation [13, 14] (Fig. 4a).

Strikingly, the *FOS*-deleted ICOs exhibited significantly elevated expression in common hepatic genes, including *ALB*, *TTR*, *CYP3A4*, *CYP2B6*, *APOC3*, *APOC1*, *SULT1A1*, and *UGT1A1*, as assessed by qRT-PCR when compared to that of the non-targeting controls (Fig. 4l). In line with an increase in *ALB* mRNA level, immunofluorescent staining demonstrated an enhanced ALB protein level in FOS-deleted-hICOs (Fig. 4m).

These results suggested deletion of *FOS* in human ICOs also boosted hepatocyte differentiation.

### *Ubr5* ablation in liver delayed hepatocyte differentiation and maturation

Among the P4 regulators, *Chaf1α* and *Ubr5* perturbation had the lowest hepatocyte regulator scores (Fig. 3f). However, as the core component of chromatin assembly factor 1 (CAF1) complex, the conditional deletion of its partner Chaf1b using Mx1-Cre (active in hematopoietic and hepatic lineages) led to postnatal lethality [45]. To facilitate the sample collection and analysis, we chose to dissect the functions of *Ubr5*, which has not been examined in the liver before. To this end, we first generated individual mICOs with *Ubr5* deletion in a similar way as illustrated in Fig. 4a and subjected them to differentiation in DM conditions. Transcriptional profiling revealed that the downregulated genes in *Ubr5* knockout mICOs were significantly associated with hepatocyte-related processes, including glucose homeostasis, fatty acid metabolic process, hormone metabolic process, xenobiotic metabolic process, and coagulation (Additional file 1: Fig. S10 and Additional file 2: Table S7), indicative of the augmented effect of *Ubr5* in hepatocyte differentiation and maturation.

To further dissect the Ubr5's role in vivo, we generated the *Ubr5* conditional knockout mice (*Ubr5*<sup>fl/fl</sup>). After crossing with Albumin-Cre mice, the exon 6 and exon 7 with flanking LoxP sites would be specifically deleted in liver cells (hepatocytes and cholangiocytes), resulting in *Ubr5* loss-of-function in the liver (Fig. 5a). Using the hepatocytes isolated by liver perfusion (Fig. 5b), we confirmed the efficient knockout of *Ubr5* by western blot analysis (Fig. 5c). The *Ubr5* knockout liver's size was comparable to *Ubr5* WT, with no notable changes in liver morphology or structure observed up to 21 days (Additional file 1: Fig. S11a). To directly evaluate the effect on hepatocyte differentiation, we sacrificed the mice at 3 weeks, and performed immunohistochemistry (IHC) analysis of several classic hepatocyte and hepatoblast markers, including Mup, Ttr, Hnf4α, and Afp. The results showed that the staining of Mup and Ttr was significantly reduced upon *Ubr5* deletion (Fig. 5d and Additional file 1: Fig. S11b). Moreover, the Albumin in the serum was reduced in *Ubr5* KO mice (Fig. 5e), suggesting the Albumin synthesis in liver was compromised.

To globally depict the gene expression profile in hepatocytes, we performed RNA-seq analysis of isolated hepatocytes from *Ubr5* WT and KO mice at 3 weeks. GO-term enrichment analysis on the 266 differentially expressed genes in *Ubr5* knockout liver revealed significant enrichment in metabolic pathways related to lipid metabolism, such as olefinic compound metabolic process, steroid metabolic process, and long-chain fatty acid metabolic process (Fig. 5f, g and Additional file 2: Table S8a, b). Indeed, indicated by Oil Red O staining, *Ubr5* knockout liver had increased lipid accumulation at 3 weeks compared to wild-type liver (Additional file 1: Fig. S11c, d), suggesting a specific role of Ubr5 in regulating the lipid metabolic pathways.

Among the 39 genes that were downregulated in *Ubr5* knockout livers, 7 genes were also significantly downregulated in *Ubr5* knockout mICOs compared to their counterpart, including several classical hepatocyte markers (*Sult5a1* and *Mup* members) (Fig. 5f and Additional file 2: Table S8c). GSEA analysis further confirmed that a set of mature hepatocyte enriched genes were downregulated in *Ubr5* KO hepatocytes (Fig. 5h and

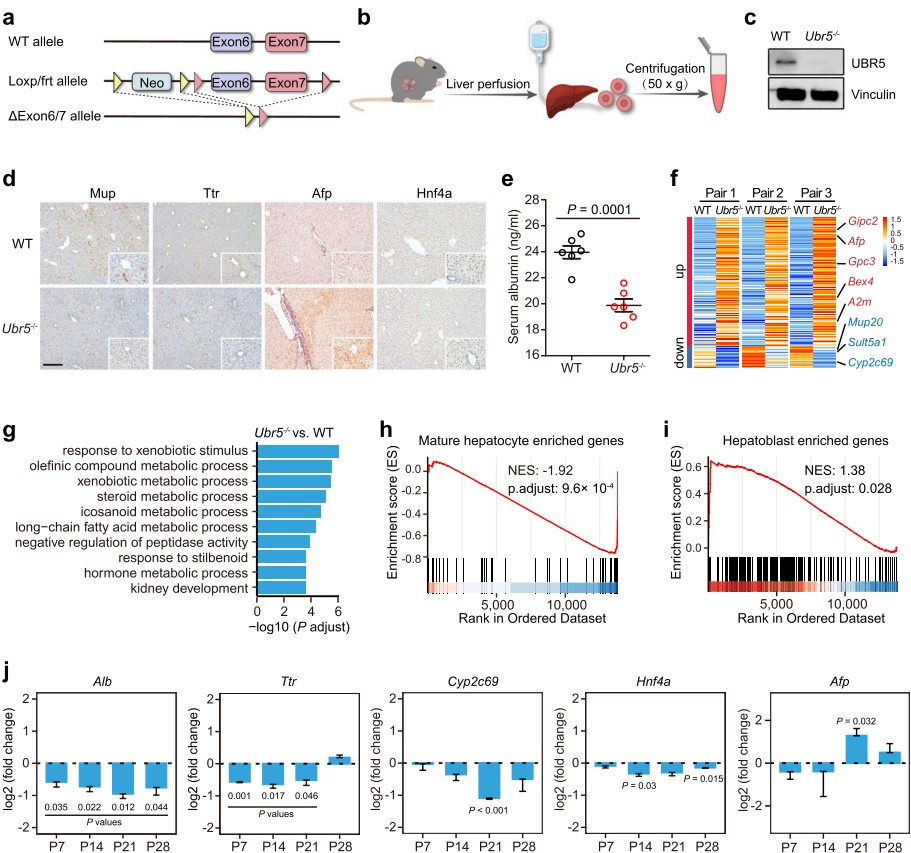

**Fig. 5** *Ubr5* ablation in liver blocks hepatocyte differentiation and maturation. **a** Schematic illustration for the generation of conditional *Ubr5* knockout mice. Exon 6 and exon 7 were deleted upon Cre-mediated recombination. **b** Schematic hepatocyte isolation by liver perfusion. **c** Western blot examination of Ubr5 protein level in WT (wild-type) and KO liver (3 weeks). Vinculin was used as a loading control. Uncropped Western blot images are provided in Additional File 1: Fig. S12. **d** Immunohistochemistry analysis of hepatic makers (Mup, Ttr, and Hnf4α) and hepatoblast maker Afp in WT and KO liver (3 weeks). Scale bars, 200 μm. Magnification = 2.5 × . **e** Serum Albumin levels in WT and *Ubr5* KO mice. Data were represented as mean ± s.e.m. (*n* = 6) and compared by two-tailed Student's *t* test, significance was set at a *P*-value less than 0.05. **f** Clustered heatmap of differentially expressed genes (|log2FoldChange| < 1 and adjusted *P*-value using BH methods < 0.05) in perfused hepatocytes from WT and *Ubr5* KO mice (*n* = 3). Selected genes were indicated in right. **g** The top 10 enriched GO-BP terms (sorted by adjusted *P*-values using the BH method) of significantly differentially expressed genes in the liver of *Ubr5* conditional KO mice compared to their wild-type counterparts. **h**, **i** Gene set enrichment analysis (GSEA) of selected gene sets encoding products related to mature hepatocyte (**h**) or hepatoblast (**i**), presented as normalized enrichment score (NES). **j** Representative hepatic gene expression in liver tissues from WT and *Ubr5* KO mice at different time point after birth. *Histone H3* was used as an internal control. The data were represented as mean ± s.e.m. (*n* = 3) and tested by one-tailed Student's *t* test. Only statistically significant comparisons (*P*-value < 0.05) were marked on the graph

Additional file 2: Table S8d), while a cohort of embryonic liver enriched genes were upregulated in *Ubr5* KO hepatocytes (Fig. 5i and Additional file 2: Table S8d). These results demonstrated that liver-specific *Ubr5* deletion led to retardation of hepatocyte differentiation and maturation. At last, to survey the time window in which Ubr5 exerts its role in hepatocyte differentiation and maturation, we isolated liver tissues from mice of different days after birth and performed qRT-PCR to detect classical hepatocyte and hepatoblast markers. Overall, *Ubr5* loss delayed differentiation during P7-P28, as indicated by the reduction of hepatocyte markers, such as *Alb*, *Ttr*, *Cyp2c69*, and

*Hnf4α* (Fig. 5j). Intriguingly, the hepatoblast marker *Afp* was increased at the time of P21 (Fig. 5j). These results suggested that *Ubr5* plays a role in postnatal development of hepatocytes in mice.

## Discussion

Single-cell CRISPR screen enables high-throughput perturbation of multiple genes with coupled functional readout of transcriptomic consequences [16–18]. Here, for the first time, we applied scCRISPR screen in a 3D organoid system to probe novel regulators for hepatic cell fate. We developed OSCAR for identifying transcriptional regulators based on their perturbation effects on the regulatory network built from the scCRISPR data. After functional validation in organoids and mice, *Fos* and *Ubr5* were demonstrated as the hepatic cell fate regulators.

Our OSCAR illustrates how scCRISPR screens in organoids could be used to reveal transcriptional regulatory mechanisms underlying in vivo development. The data analysis framework provides a methodology that is readily transferable to other systems for identification of determinants of complex biological processes. With the success of organoid-based application, we plan to apply OSCAR in vivo for further dissecting genetic circuits of liver development, regeneration, and tumorigenesis. While the delivery of sgRNAs to the fetal liver will be challenging, this might be overcome through recently reported transgenic inducible mosaic sgRNA system [46].

Technically, one major advance of our study is applying the scCRISPR screen in the organoid system. Different from 2D cell culture, 3D organoid recapitulate the in vivo tissue architecture and behavior. Compared with the iPSC system, the cholangiocyte organoid differentiation system offers a more convenient one-step differentiation protocol and higher genomic stability during passaging [13], providing a more easily manipulable system for screening. However, previous CRISPR screens in organoids relied on cell proliferation as readout [47–51], which would limit the ability to map regulators in complex processes, such as differentiation, maturation, and aging. scCRISPR screen connects genes with complex phenotypes using the whole transcriptome for every single cell as the readout, which is particularly useful for the heterogeneous organoid system. To study the neural fate regulators in brain development, Fleck et al. perturbed 20 TFs in iPSCs and generated mosaic neural organoids from them. The regulators of dorsal telencephalon commitment were identified based on the enrichment of sgRNAs in different cell clusters in the scRNA-seq data [52].

In our in-organoid scCRISPR screen for hepatocyte fate regulators, we developed a strategy to identify the key regulators from the scCRISPR screen data based on the changes in regulatory networks. Compared to previous methods, the OSCAR framework has some advantages. First, using activities of TF-centered gene modules derived from the screen data, we were able to alleviate the impact of inherent noise in scCRISPR screens and thus reduced the cell number required for robustly measuring perturbation effects. Second, we were able to dissect different functional pathways in the complex processes and link them to perturbed genes. We obtained modules that are directly associated with hepatocyte function, such as those regulating the expression of hepatocyte enzymes in lipid and glucose metabolism. Even though the study was performed in organoids in vitro, the gene modules recapitulated part of the regulatory network in the in vivo development of mouse

liver. Hence, our OSCAR provided a rapid, reliable analysis method for scCRISPR screens, which can be applied to study regulatory programs in other complex biological processes.

After identifying key regulators based on their overall regulatory profiles, we separated the perturbed and unperturbed cells based on their gene expression patterns, and removed the unperturbed cells for each sgRNA to ensure the accurate mapping of perturbation effects from a heterogeneous cell population in organoids. We were able to rank the regulators based on their direct impacts on the expression of hepatocyte marker genes. We validated the functions of the top-ranked positive and negative regulators. *Fos* knockout directly promoted hepatocyte differentiation in organoids. Not only known as an oncogene, *Fos* was also reported as a fate regulator in multiple lineages. Constitutive *Fos* deletion led to a differentiation block of osteoclasts and lineage transition to macrophages [53]. Overexpression of Fos in vitro inhibited the differentiation of chondrocytes [54]. We discovered a specific role of *Fos* in hepatocyte differentiation and maturation, as loss of *Fos* significantly activated hepatic metabolic pathways and functions in both human and mouse ICO-derived hepatocyte-like cells. Consistent with our observation, conditional knockout of *Fos* in mouse liver led to the upregulation of metabolic pathways, and vice versa [44]. The consistency of FOS protein function in mouse and human also reveals the power for performing genetic screen in a model with uniform genetic background. Interestingly, *FOS* was also reported as the direct target of Hippo-Yap signaling, and inhibition of *FOS* activity with small molecule T5524 would rescue the liver organ overgrowth induced by *Yap* [55]. Considering our data, *FOS* would possibly be a potential druggable gene to rewire the metabolic status of dedifferentiated hepatocytes in cancer.

Another regulator we functionally validated was *Ubr5*, constitutive knockout of *Ubr5* in mice was embryonic lethal [56], while conditional disruption of the Ubr5 catalytical domain in early B-lymphocytes impairs the B cell maturation, indicating the role of *Ubr5* in cell fate regulation [57]. Our findings that Ubr5 ablation delayed hepatocyte differentiation and maturation revealed the significance of this regulator in liver development. The transcriptional profiling revealed a strong defect of lipid metabolism, which was further supported by the increased lipid accumulation observed in *Ubr5* knockout liver at 3 weeks. The potential molecular mechanism underlying Ubr5's regulation on hepatocyte differentiation needs further investigations. Given that Ubr5 is an E3 ligase, the finding of Ubr5's substrates and its interactions with other proteins might provide more insights.

Our screen was conducted using ICOs, which possess a cholangiocyte identity and could be converted to hepatocyte-like cells upon induction. The regulators we identified in the screen are factors that are involved in the cholangiocyte-hepatocyte identity switch. While we have validated the role of *Ubr5* in mouse liver development, additional studies employing the iPSC-to-hepatocyte differentiation system are crucial to ascertain these factors' role in human hepatocyte differentiation and maturation.

## Conclusions

In summary, we developed OSCAR, a framework using regulon activities as readouts to dissect gene knockout effects in organoids. By integrating scCRISPR and liver organoid system, we achieved pooled library screening in organoids and identified novel regulators that significantly impact hepatocyte cell fate. Our results demonstrate that OSCAR is a powerful and scalable method to discover new regulators of cell fate determination.

## Methods

### Human material for organoid culture

Human liver tissue was obtained from Zhongshan Hospital Fudan University. Informed consents were obtained from all the patients. All procedures were in accordance with the ethical standards of the Medical Ethical Council of Zhongshan Hospital Fudan University.

### Mouse models

*Rosa26-LSL-SpCas9* and *Alb-Cre* mice were obtained from Jackson Laboratory. *Rosa26-SpCas9* was generated by crossing *Rosa26-LSL-SpCas9* with *Dppa3-Cre* obtained from Shanghai Model Organisms. Ubr5 fl/fl mice were generated by Model Animal Research Center of Nanjing University (MARC, Nanjing, China). The targeting strategy is shown in Fig. 5a. Genotyping primers are listed in Additional file 2: Table S9a.

All mice were maintained in C57BL/6 background. All breeding and experimental procedures were performed in accordance with the relevant guidelines and regulations and with the approval of the Institutional Animal Care and Use Committee at Fudan University.

### Isolation, culture, and differentiation of Cas9-expressing mouse intrahepatic cholangiocyte organoids (mICOs)

To isolate mouse bile-duct organoid, livers from *Rosa26-SpCas9* knockin mice were removed and minced into small pieces of roughly 0.5 mm$^3$. The minced tissue was washed three times with ice-cold wash medium to remove the red blood cells, and then incubated with digestion buffer containing 0.125 mg/ml collagenase, 0.125 mg/ml dispase II, and 0.1 mg/ml DNase I at 37 °C for 45 min with agitation. The cell suspension was filtered through a 100-μm cell strainer to exclude the tissue residue and washed with basal medium for 4 times to remove hepatocytes from the suspension. The resulting pellets were resuspended in 300 μL of ice-cold Matrigel and allowed to seed into six wells of a 24-well plate. After polymerizing, the Matrigel droplets were overlaid with growth medium as previously described [14].

To passage the organoids, Matrigel domes in medium were scraped mechanically and pipetted up and down using a 1000-μL pipette. The organoid suspension was transferred to a 15-ml centrifuge tube supplemented with cold basal medium and pipetted up and down several times to remove the Matrigel from the organoids. After centrifuging at $200 \times g$ for 3 min at 8 °C, the organoids were resuspended in 500 μL basal medium, transferred to a 1.5-ml centrifuge tube, and pipetted up and down using a 200-μL pipette to disrupt the organoids completely into fragments. The cell fragments were collected by centrifuge at $200 \times g$ for 3 min and resuspended in fresh Matrigel before plating into the wells of a 24-well plate. The organoids were passaged every 5–7 days as desired.

To differentiate the cells within the organoids to hepatocyte-like cells, the organoids seeded as above described were cultured in expansion medium for 2 days and then transferred into differentiation medium (day 0) which was changed each day for up to

4 days (day 4) for CROP-seq experiment or 9 days (day 12) for individual validation experiment. From day 4 to day 7 for CROP-seq experiment, 3 μM dexamethasone was added into the differentiation medium which was replaced each day until the differentiation endpoint.

### CROP-seq library construction, lentiviral packing

A CROP-seq library with 246 sgRNAs targeting 78 transcription factors or chromatin remodeling factors (3 sgRNAs per gene except for *Hnf4α* with 5 sgRNAs, 236 sgRNAs in total) to be potentially involved in liver development and regeneration and negative controls (10 non-targeting sgRNAs) was constructed. Briefly, oligos were synthesized, annealed, and cloned individually into the CROP-seq-Guide-mCherry vector (modified from CROP-seq-Guide-Puro, Addgene plasmid #86,708). The plasmids were verified by Sanger sequencing and pooled together for lentiviral packing.

For lentiviral packing, the CROP-seq library plasmids together with the helper plasmids psPAX2 (Addgene plasmid #12,260) and pMD2.G (Addgene plasmid #12,259) were transfected into HEK293T cells using Neofect transfection reagent (Neo biotech). At 24 h after transfection, medium was changed with viral producing medium (Lonza). Sixty hours post transfection, lentivirus particles were collected, filtered through a 0.45-μm filter, concentrated by centrifuging at $120,000 \times g$ for 3 h, aliquoted and frozen at $-80$ °C.

### CROP-seq screen

For CROP-seq screening, Matrigel beds should be prepared in advance as previously described [33]. Briefly, Matrigel was diluted in expansion medium to a concentration of 25% and plated evenly to 24-well plates. These beds were placed in 5% $CO_2$ incubator at 37 °C to be solidified overnight. BEC-organoids were enzymatically dissociated into small cell clusters ($\sim 5$ cells for each cluster) with prewarmed TrypLE supplemented with 1 mM EDTA (pH 8.0) for 10 min. To achieve $\sim 2000 \times$ coverage of the library, 500,000 cell clusters were infected with CROP-seq library lentivirus (MOI $\sim 0.3$) in expansion medium containing 10% Matrigel, 8 μg/ml polybrene, and 10 μM ROCK inhibitor (Y-27632). The suspension of cell clusters and lentivirus were added dropwise onto the Matrigel beds (10,000 clusters for each Matrigel bed). At 24-h post-infection, organoids were scraped, washed, and reseeded into the 24-well plates for expansion. Three days after infection, organoids were dissociated with prewarmed TrypLE supplemented with 1 mM EDTA (pH 8.0) for 20 min to obtain single-cell suspension. The GFP/mCherry double positive single cells were enriched by FACS sorting and seeded into wells of 24-well plate. After expanding for 5 days, the organoids were passaged, expanded for 2 days, and subject to differentiation medium (DM condition) for differentiation as described above.

At the endpoint of differentiation, approximately 30,000 BECs from EM condition and 60,000 differentiated hepatocyte-like cells from DM condition were captured by the Chromium Controller using Chromium Next GEM Single Cell 3' kit v2 (10 × Genomics) with 10,000 input cells for each lane. Libraries were built according to the manufacturer's instructions and sequenced on an Illumina NovaSeq 6000 platform.

### Analysis of single sgRNA perturbation effect using scMAGeCK

We analyzed the alteration of gene expression for each sgRNA by the linear regression method (LR) of scMAGeCK R package (version 1.6.0) running with default parameters. The sign and value of obtained coefficients matrix could reflect the changing direction and degree of gene expression phenotypes respectively in the presence of perturbations.

### The OSCAR framework for data analysis

#### *CROP-seq raw data processing*

Single-cell sequencing data from each library was processed using "cellranger count" pipeline in CellRanger (version 3.1.0) suite with default parameters to generate an individual expression matrix. The feature-barcode matrix was imported in R (version 4.0.3) and analyzed with the Seurat R package (version 3.2.3). mm10 was used as the reference genome. We retained cells with more than 200 expressed genes and less than 6000 expressed genes as well as less than 10% mitochondrial genes. For the assignment of sgRNAs to cells, a "cropseq_count.py" script [23] was used to collect cell identity information from bam files generated by Cell Ranger. Cells with unique sgRNA were retained for downstream analysis.

#### *Regulon prediction and activity assessment*

pySCENIC (version 1.2.4) was used to perform single-cell regulatory network inference. According to the published protocol [58], the gene count matrix was loaded into SCENIC with gene symbols as row names, and cell barcode as column names. To filter low-quality genes, we only kept genes with at least 6 UMI counts across all samples and detected in at least 1% of cells. Then the co-expression network was built with GRNBoost, followed by pruned by Rcistarget to generate gene regulatory network. Finally, the regulon activity for each cell was scored according to AUCell.

#### *Construction of regulatory maps and identification of regulators*

In order to compare the regulatory pattern among perturbations, we first filtered sgRNAs with less than 40 cells supported. The effect size of a perturbation on a regulon was obtained via the mean regulon activity scores of each target gene group minus the mean regulon activity scores of non-target cells. The Spearman's correlation coefficient was calculated for each perturbation pair or regulon pair. The perturbation modules ($P_i$) and regulon modules ($R_j$) were identified based on $k$-means clustering via the function "kmeans."

We constructed the regulatory map of modules ($P_i$-$R_j$) by performing hypergeometric tests, following the strategy of IreNA [59]. The regulatory relationships between perturbation and regulon modules ($P_i$-$R_j$) were assessed by reshaping and simplifying the effect size matrix of perturbation and regulon. Firstly, we retained the regulation pairs filtered by Wilcoxon rank sum test ($p < 0.05$), and divided them into active (effect size $> 0$) and repressive (effect size $< 0$) regulations. Secondly, we applied the following

formula to compute the statistical significance (*P*-value) from a perturbation module *i* ($P_i$) to a regulon module *j* ($R_j$):

$$P(x = k) = \frac{\binom{K_P}{k}\binom{N-K_P}{K_R-k}}{\binom{N}{K_R}} \tag{1}$$

where *N*, $K_P$, $K_R$, and *k* denote the total number of regulations in the matrix, the number of regulations derived from $P_i$, the number of regulations targeting $R_j$, and the number of regulations targeting $R_j$ from $P_i$, respectively. The *P*-value was adjusted by Benjamini–Hochberg method and the threshold value was set to 0.05. Thirdly, we utilized Cytoscape (version 3.7.2) to visualize the whole regulatory map of modules ($P_i$-$R_j$). The reliability of the module regulatory relationship is measured by the logarithm of adjusted *P*-value ($-\log_{10}$ adjusted *P*-value), and we only retained the mapped lines with adjusted *P*-value less than 0.05.

### Refine sgRNA assignments by identification of unperturbed cells

Due to the variability of editing outcomes for CRISPR-Cas9 system, the presence of a sgRNA in a cell does not always result in a loss-of-function of its target gene. To accurately map the perturbation effects of selected sgRNAs, we exploited MIMOSCA [18, 52] to calculate the perturbation probability for each cell, using the default parameters except that parameter "l1_ratio" was set to 0.5. Theoretically, the distribution curve of perturbation probability will be bimodal, consisting of a population of cells with no perturbation (either unedited or edited but has eligible effects on protein function) and a population of cells with perturbation. We utilized Gaussian mixture models (GMM) from mixtools R package (version 1.2.0) to separate cells into two groups ($k=2$) for each sgRNA. If the probability of a candidate cell entering the perturbed group is less than 0.05, the cell will be regarded as unperturbed cell and removed for further analysis.

### Scoring of hepatocyte markers

To compare and rank the effects of candidate perturbations on our focused hepatocyte markers (*Alb*, *Ttr*, *Mup20*, *Sult1a1*, *Cyp3a13*, *Abcc3*, *Tff3*, *Trf*, and *Fga*), we defined the following formula to compute a score for each perturbation:

$$Score_i = \sum_{j=1}^{k} log_2 FC_{ij}/k, \quad i = 1, 2, \ldots, n \tag{2}$$

where score$_i$ donates the composite score of perturbation *i*, *k* donates the total number of markers, *n* donates the total number of perturbations, and FC$_{ij}$ donates the fold change of normalized expression of marker *j* in cells with perturbation *i* compared with unperturbed cells.

### Comparison of OSCAR with scMAGeCK and MIMOSCA

scMAGeCK (LR) and MIMOSCA were run with default parameters using the expression matrix and cell identities as the input. The output matrices of the perturbation effects were used for the comparison, using the metrics described below.

### Consistency

To compare the consistency of the three methods, a relative perturbation effect of each target was calculated as Spearman's correlation coefficient of the top 200 variable features between that target and *Hnf4α*. The consistency was then evaluated by pairwise Spearman's correlation of the relative perturbation effects obtained by three methods.

### Robustness

Since the coefficients in the perturbation effect matrix obtained by the three methods were close to 0 for most perturbations, the agreement on the sign of the coefficient was used to evaluate the robustness instead of the value of the coefficient. Seven target genes with perturbed cell numbers larger than 600 were included in the analysis.

(I)　Firstly, we construct a cell-TF matrix (defined as $M_0 = (A_{ij})_{n \times m}$) with $m$ cells and $n$ TFs that are recognized both as the core members of regulons computed by SCENIC and the 4000 highly variable genes computed by Seurat, where $m \geq j$, $n \geq i$, $A_{ij}$ refers to the normalized regulon activity or TF expression according to specified method $F$ among methods to be evaluated. In addition, the target identity vector of $m$ cells is defined as $P_0$ with $q$ groups. Taking the cell-TF matrix $M_0$ and the identity vector $P_0$ as inputs to $F$, the output was the coefficient matrix $C_0 = (V_{ij})_{n \times q}$ with $q$ targets and $n$ TFs, which converted to $C_0^* = (V_{ij}^*)_{n \times q}$ via the "sign" function in R, where $q \geq j$, $n \geq i$, $V_{ij}$ refers to the coefficient and $V_{ij}^*$ refers to the sign of the coefficient.

(II)　For each target group, at a given cell number $N$, $P_0$ was downsampled 100 times, and $P_t$ was defined as the vector obtained at the $t$ time. For $t = 1, 2, \ldots, 100$, a Matrix $M_t$ was obtained by extracting cells from $M_0$, according to cell identities in $P_t$. We then derived the matrix $C_t^*$ from $P_t$ and $M_t$ using the same procedure as in (I).

(III)　We obtained the co-directionality matrix $D = (R_{ij})_{n \times q}$ with $q$ targets and $n$ TFs, where $q \geq j$, $n \geq i$. Here, $R_{ij}$ was resigned to reflect the co-directionality between 100 downsamples and population, and computed as below:

$$R_{ij} = \sum_{t=1}^{100} I(C_{tij}^* = C_{0ij}^*)/100$$

where $I$, $t$, $i$, and $j$ denote the indicator function, the downsample round number, TF $i$, and target $j$, respectively. To measure the robustness of target $j$, we have defined an indicator named "robustness score" or *RS for each target j within q targets*:

$$RS_j = ||D_j - [1]||_2, \quad j = 1, 2, \ldots, q$$

### GO terms

The TFs were first ranked by the variance of their coefficients in each method. The functional GO terms enriched for the selected number of the top-ranked TFs were clustered

and the proportion of functional modules that matched the prior knowledge were compared among three methods.

### Bulk RNA-seq analysis

Total RNA was extracted with RNAprep pure micro kit (Tiangen). RNA amount and integrity were measured by Bioanalyzer 2100 (Agilent). Libraries for RNA sequencing were generated using the Illumina TruSeq RNA Sample Prep kit v2 and sequenced on an Illumina NovaSeq 6000 platform. Reads were processed using the following pipeline. First, the quality of raw sequencing data was checked by FastQC (version 0.11.8) and MultiQC (version 1.6). Then STAR (version 2.7.1a) was used to align reads to mouse GRCm38 genome with parameters "–outSAMtype BAM SortedByCoordinate –outSAMunmapped Within –outSAMattributes Standard." Bam files were sorted and indexed by samtools (version 1.15.1), and count matrices were generated by feature-Counts (version 2.0.1). Downstream analysis was completed in R (version 4.0.3). After performing the differential expression test by DESeq2 (version 1.30.0), upregulated genes (log2 (fold change) $\geq$ 1 and p.adjust < 0.05) and downregulated genes (log2 (fold change) $\leq$ −1 and p.adjust < 0.05) were identified. Heatmaps was generated by pheatmap (version 1.0.12).

### Enrichment analysis

Functional enrichment analysis for specified genes was performed by the function "enrichGO" from ClusterProfiler (version 3.18.0) with parameters "ont='BP', pvalueCutoff=0.05, qvalueCutoff=0.2." In order to concisely comprehend the enrichment terms of target genes paird with top regulons, the redundancy of enrichment items was reduced by the function "simplify" with parameters "cutoff=0.7, measure="Wang"" and the term clusters based on semantic similarity were obtained by the method "binary cut" provided by simplifyEnrichment (version 1.4.0). Sankey map generated by ggalluvial (version 0.12.3) was applicable to the visualization of relationships between regulons and various functional clusters. Gene set enrichment analysis was implemented by the function "GSEA" from clusterProfiler and "gseaplot2" from enrichplot (version 1.14.2).

### Pseudotime analysis

The dynamic changes in single-cell profiles of specified genes were assessed by Monocle2 (version 2.22.0) [60]. A continuous pseudotime heatmap was generated and smoothed by the function "plot_pseudotime_heatmap" to visually compare the temporal patterns of different genes.

### qRT-PCR

Total RNA was extracted with the rNeasy Mini Kit (Qiagen) according to the manufacturer's instructions. Complementary DNA was synthesized with the GoScript Reverse Transcription System (Promega). qRT-PCR reactions were performed with SYBR qPCR Mix (Biomake) in triplicates on the CFX96 Touch System (Bio-Rad). Primer pairs are listed in Additional file 2: Table S9b.

### Immunofluorescence and ELISA

The organoids were collected and washed with cold PBS after the medium was discarded, pelleted by centrifugation (1 min at $300 \times g$), then fixed in 4% paraformaldehyde for 15 min. The fixed organoids were washed with PBS in tube three times, permeabilized with 0.25% Triton X-100 for 15 min, then blocked in PBST solution (0.1% Triton X-100) containing 1% donkey serum (Solarbio) for 1 h at room temperature. The samples were then incubated overnight with primary antibody at 4 °C. Fluorescein-labeled secondary antibodies (Thermo Fisher Scientific, 1:200) and 4,6-diamidino-2-phenylindole (DAPI) were applied for 1 h at room temperature. Confocal laser scanning was done on an Olympus FV3000 laser-scanning microscope. Albumin ELISA was performed according to the manufacturer's instructions (Abcam, ab108792).

### Oil Red O staining and immunohistochemistry

Liver tissues were fixed with 4% paraformaldehyde and embedded in optimal cutting temperature (OCT) compound or paraffin. For Oil Red O staining, working solution was prepared from Oil Red O stock solution mixed 3:2 with water and incubated at 4 °C overnight. Solution was filtered through 0.45-μm filters and applied on OCT-embedded liver sections for 10 min, followed by incubating in 60% (v/v) isopropyl alcohol for 3 s. Slides were washed twice in water, and counterstained with hematoxylin. Lipid droplets was quantified using the Image-Pro Plus software by measuring area occupied by red pixels. For immunohistochemistry, paraffin-embedded liver sections were deparaffinized in xylene and graded alcohols, followed by antigen retrieval, and endogenous peroxidase quenched by $H_2O_2$. Sections were then blocked with 1% BSA in PBS for 30 min, and incubated overnight at 4 °C with α-Mup (Santa Cruz, 1:200), α-Ttr (ProteinTech, 1:100), α-HNF4α (Abcam, 1:500), and α-AFP (ProteinTech, 1:100). Secondary biotinylated anti-rabbit antibody (Vector Labs, 1:400) was added for 2 h at room temperature, followed by detection with streptavidin-HRP (Vector Labs) and DAB + chromogen (Vector Labs) according to the manufacturer's recommendations. Slides were counterstained with Mayer's hematoxylin, dehydrated, and mounted with Eukitt (Sigma). Images were taken by Vectra Automated Quantitative Pathology Imaging System (Perkin Elmer). Image-Pro Plus version 7.0 software was used to access the integrated optical density (IOD) value of the IHC sections. The signal density of tissue areas from five randomly selected fields were counted in a blinded manner and subjected to statistical analysis.

### Immunoblotting

These assays were performed as previously described. The following antibodies were used: α-UBR5 (Bethyl, 1:2,000), α-FOS (CST, 1:2,000), α-Vinculin (Abcam, 1:4,000), α-β-actin (Cell signaling technology, 1:2000), and HRP-conjugated α-mouse IgG and α-rabbit IgG (Epizyme, 1:10,000).

### Fluorescence activated cell sorting (FACS)

For organoids, single-cell suspension was dissociated by 1xTrypLE (Gibco) containing 15 U/ml DNase at 37 ℃ for 15–20 min. To avoid cell clumps, suspend the cells with pipet every 5 min. Flow cytometry was performed using FACSAria II (BD) flow cytometer to sort mCherry$^+$ cells.

### Organoid growth assessment

For assessing mICO growth, single-cell suspension dissociated by 1xTrypLE (Gibco) were seeded in wells of a 48-well plate (2000 cells per well). Organoid growth was assessed using the CellTiter-Glo 3D Cell Viability Assay (Promega) according to the manufacturer's instructions.

### Statistical analysis

The hypothesis tests in this paper were carried out using the 'stat' package function in R language (version 4.0.2) or Prism Software (version 6.0). For small sample dataset that did not pass the Shapiro–Wilk test ($P$-value < 0.05), logarithmic transformations were applied to enhance data normality. Student's $t$ test was employed for comparisons between two groups, while one-way analysis of variance (ANOVA) were applied for comparisons involving more than two groups. Spearman's correlation analysis was used to quantify the correlation among datasets. The significance threshold of the $P$-value was set to 0.05 by default. Adjustments for multiple hypothesis testing were made using either the false discovery rate (FDR; achieved by Benjamini–Hochberg or $q$-value) or the family-wise error rate (FWER; achieved by Bonferroni), as specified in the manuscript.

### Review history

The review history is available as Additional file 3.

### Supplementary Information

---

**Additional file 1: Fig. S1.** Comparison of different differentiation strategies for mICOs. mICOs isolated from livers of spCas9-EGFP knock-in mice were cultured under DM for differentiation or EM for expansion for 7 or 12 days. Cultures were harvested for transcriptional profiling. a qRT-PCR analysis showing relative gene expression as mean ± s.e.m. (*n* = 4) of known hepatocyte markers (Alb, Ttr, Cyp3a11, Apoa1, Mup20, Mrp2, Sutl1a1, and Aldh1a1) or biliary duct markers (Sox9 and Spp1) for mICO cultures maintained under expansion medium (EM) or transferred to differentiation medium (DM) for 7 or 12 days (DM_7 or DM_12). Following one-way ANOVA, pairwise comparisons were performed using the Tukey-HSD test. Only statistically significant comparisons (*P*-value <0.05) were marked. b Principal-component analysis (PCA) of transcriptional profiles of the samples under each condition. X-axis: the principal component with the largest explanatory variance. Y-axis: the principal component with the second largest explanatory variance. c Venn diagram showing the overlap of upregulated and downregulated genes in DM_7 and DM_12 groups compared to the EM group. d The top 5 gene ontology biological processes enriched for upregulated or downregulated genes shared in DM_7 and DM_12 groups. e GSEA of transcriptional profile using KEGG gene sets of MsigDB. Bubble size indicates -log10 (FDR q-value) and the color of bubble denotes normalized enrichment score (NES). f-h Enrichment score (ES) plots displaying the top 3 pathways of the most highly enriched gene sets including retinol metabolism (f), drug metabolism cytochrome p450 (g), and complement and coagulation cascades (h). The statistically significance threshold was set to 0.05 and the adjusted method used was the q-value. **Fig. S2.** sgRNA assignments and scRNA-seq quality control in pilot CROP-seq screen. a The distribution of the number of sgRNAs detected per cell in EM and DM group, respectively. b Violin plots of the number of genes (feature), number of UMIs (count) and mitochondrial gene percentages for the cells with unique sgRNA assignments in EM or DM group. Cells with more than 200 expressed genes and less than 6,000 expressed genes as well as less than 10% mitochondrial genes were retained. c The number of cells expressing a unique sgRNA targeting the same gene. **Fig. S3.** sgRNA assignments and scRNA-seq quality control in CROP-seq screen. a Distribution of the number of sgRNAs detected per cell in EM and DM groups, respectively. b Quality controls of CROP-seq screen, with the same criteria as that in the pilot study. c Number of single cells expressing a unique sgRNA for each targeting gene. d

Distribution of cell numbers for each perturbation in EM (left) and DM (right) groups. **Fig. S4.** Sankey plot of enrichment analysis for the top 20 variable regulons. The first column: Top 20 variable regulons (number of terms > 0). The second column: clusters of GO-BP terms (adjusted *P*-value using BH methods < 0.05). The third column: the focused terms. The edge linking the first column to the second column: the term i of regulon j is the member of cluster k. The thickness of edge: the number of terms. **Fig. S5.** Evaluating the number of cells required for robustly measuring the regulon activities and expression. a-i Scatter plot showing the Spearman's correlations between average regulon activities (blue points) or average gene expression (red points) in the indicated number of cells and that in the whole sample. The cells were randomly sampled 1000 times for each indicated cell number. The curves represent the median point of the 1000-time iterations. Cells with sgRNAs targeting Six5 (a), Tead2 (b), Tet2 (c), Spic (d), Relb (e), Dnmt3b (f and g), Jdp2 (h), Onecut1 (i) were shown. **Fig. S6.** Comparison of the three methods on the performance of mapping perturbation effects. a-c Scatter plots showing the correlation of perturbation effects among the three methods. Each point represents the perturbation effect of a target relative to Hnf4a, which is calculated as the Spearman's correlation coefficient between the top 200 variable features for that target and those for Hnf4a. d The line plot shows the robustness of perturbation effects identified by the three methods by subsampling different numbers of cells. The robustness score represents the agreement of coefficients calculated from the 100 subsets of data with those calculated from the whole dataset. e, f The line plot shows the proportion of the key GO term module associated with different numbers of top variable TFs identified by the three methods, using our OSCAR dataset (e) or the ESC CROP-seq data (f). **Fig. S7.** Association of each perturbed gene with the activities of top 20 variable regulons in each regulon program. Column: each perturbed gene from P1-P4 perturbation groups. Row: top 20 variable regulons for each regulon program. **Fig. S8.** Filtering unperturbed cells from scRNA-seq data by MIMOSCA and Gaussian fitting. a Distribution of perturbation probability of each sgRNA before filtering. The innermost layer: hierarchical clustering tree indicating the similarity of the distribution of perturbation probability calculated by MIMOSCA. The second layer: annular heatmap showing the relative density ( , where $Count_i$ denotes the number of cells within $bin_i$, and the width of each bin is 0.05) of each sgRNA. The third layer: annular bar plot showing the number of cells before filtering. As a reference, the number of cells with sgRNA Sfpi1−1 is 183. The fourth layer: annular band plot showing the performance of GMM. From inside to outside of the fourth layer, the location of bands indicates the probability of each cell being assigned to the second cluster (Higher probability of perturbation) of the GMM in each bin, and the color of bands indicates the probability of perturbation of each cell. b, c The angle of the polar coordinate plot represents the proportion of cells retained, and the distance from the pole represents the number of cells retained (log2 transformation). Histogram showing the distribution of the proportion of cells retained for Module P2 (b) or Module P4 (c). **Fig. S9.** Comparison of growth and gene expression in Fos-KO and control mICOs under expansion or differentiation medium. a Representative images showing cyst structures of mICOs stably expressing either non-targeting sgRNA (sgNT) or sgRNA targeting Fos (sgFos_1, sgFos_2, sgFos_3). Scale bars: 500 μm. b Upper panel: Growth of three Fos-KO lines and control mICOs in EM condition was monitored using the Cell-titer Glo assay. Luminescence values were taken every 24 hours. Error bars indicate the standard error of the mean. Lower panel: Growth rates for the Fos-KO lines and control mICOs in expansion medium (EM) were determined over 24-hour intervals across a 6-day period. Growth rates were calculated by dividing each day's luminescence value by the previous day's average. Error bars represent standard errors of the mean (n = 3). Statistical significance was tested using one-way ANOVA, followed by Bonferroni's correction for pre-planned pairwise comparisons between sgNT and each sgFos line. A *P*-value less than 0.05 was considered statistically significant, and only such comparisons are marked on the graph. c qRT-PCR analysis of the relative gene expression of selected cholangiocyte markers (Sox9 and Spp1) and hepatic markers (Alb, Ttr, Apoa1, Mup20, and Mrp2) in mICO cultures. Organoid cultures were maintained under expansion medium (EM) or switched to differentiation medium (DM) for 7 days. Statistical significance was assessed using one-way ANOVA, followed by Bonferroni's correction for pre-planned pairwise comparisons between sgNT and sgFos_3 under both EM and DM conditions. A *P*-value less than 0.05 was considered statistically significant. **Fig. S10.** Ubr5 depletion in mICOs weakens hepatocyte differentiation and maturation. Ubr5 KO mICOs and the NT controls were maintained under DM condition for differentiation with the strategy as illustrated in Fig. 3a. a Volcano plot shows differentially expressed genes (|log2FoldChange| < 1 and an adjusted P-value using the BH methods < 0.05). Blue, downregulated genes; red, upregulated genes. Representative markers were labelled. b Selected GO terms significantly enriched for genes down-regulated in Ubr5 KO cultures. c-g Heatmaps showing differentially expressed genes involved in glucose homeostasis (c), fatty acid metabolic process (d), hormone metabolic process (e), xenobiotic metabolic process (f) and coagulation (g). **Fig. S11.** Ubr5 depletion in vivo weakens hepatocyte differentiation and maturation. a Morphology of WT and Ubr5-KO livers from mice at P21. Scale bars: 100 μm. b Quantification of IHC images by calculating the relative Integrated Optical Density (IOD). Data are represented as mean ± s.e.m. (n = 5) and were compared by the two-tailed student's t test. c Representative images of Oil Red O staining of WT and Ubr5-KO livers from two pairs of mice at P21. Scale bars, 100 μm. d Percentages of lipid droplets area in (c) were shown as mean ± s.e.m. (n = 4) and compared by Wilcoxon rank sum test within each pair. A *P*-value less than 0.05 was considered statistically significant. **Fig. S12.** Uncropped images of Western blots. a Western blot gel images for FOS and β-actin; boxes indicate the cropped regions in Fig. 4b. b Images of western blot gel for UBR5 and Vinculin; the cropped regions in Fig. 5c are indicated by the boxes.

**Additional file 2: Table S1.** Downstream analysis of RNA-seq data for organoids at different culturing conditions (EM, DM_7 and DM_12). Related to Fig. S1c, d. **Table S2.** sgRNA sequences and their targeting genes used in the pilot screen. **Table S3.** sgRNA sequences and their targeting genes used in the OSCAR screen. **Table S4.** Detailed data of regulons computed by SCENIC. **Table S5.** Enriched GO-BP terms of top 20 variable regulon (Terms > 0). Related to Fig. S4. **Table S6.** Downstream analysis of RNA-seq data for organoids at different culturing conditions (Fos KO vs WT). Related to Fig. 4c-f. **Table S7.** Downstream analysis of RNA-seq data for organoids at different culturing conditions (Ubr5 KO vs WT). Related to Fig. S9. **Table S8.** Downstream analysis of RNA-seq data for hepatocytes from Ubr5 conditional knockout mice and the wildtype mice (Ubr5 KO vs WT). Related to Fig. 5f-i. **Table S9.**

The information of primers used in this study. **Table S10.** The source data and detailed statistics associated with Figs. 4 and 5, Fig. S1, Fig. S9 and Fig. S11.

**Additional file 3.** Review history.

## Acknowledgements
We thank Dr. Chunyan Liu at Nanjing University for helpful discussions. We acknowledge the use of High-performance Computing Platform at the Center for Bioinformatics, Institute of Basic Medical Sciences Chinese Academy of Medical Sciences, School of Basic Medicine Peking Union Medical College.

## Peer review information

## Authors' contributions
X.W., B.Z. and J.L. conceived and designed the project. J.L., J.W., X.L., D.W. and Y.G. performed experiments and contributed to data processing. J.C., J.Q. and X.W. performed bioinformatics analyses. X.W. and B.Z. supervised the project. J.L., J.W., J.C. and X.W. wrote the manuscript. L.D. and J.Y. provided valuable discussion.

## Funding
This work was granted by National Key Research and Development Program of China (2018YFA0109800 to X.W.), National Natural Science Foundation of China (32122023 and 32070603 to X.W., 32022022 and 31970761 to B.Z., 82172887 to J.L.), CAMS Innovation Fund for Medical Sciences (CIMS) (2021-I2M-1–066 to J.L.), National High Level Hospital Clinical Research Funding (2023-PUMCH-E-008 to X.W.), the Key Research and Development Program of Yunnan Province (202302AA310024 to B.Z.) and State Key Laboratory Special Fund (2060204 to X.W.).

## Availability of data and materials
The omics data from this study have been deposited in the National Genomics Data Center (NGDC) under the accession number NGDC: PRJCA014442. This includes both bulk RNA-seq data (https://ngdc.cncb.ac.cn/gsa/browse/CRA009688) [61] and scRNA-seq data (https://ngdc.cncb.ac.cn/gsa/browse/CRA009621) [62].
The code for reproducing the main analyses of the paper is available under the MIT license at GitHub (https://github.com/Wangxiaoyue-lab/OSCAR) [63] and Zenodo (https://doi.org/10.5281/zenodo.8385064) [64].

# Declarations

## Ethics approval and consent to participate
This study was conducted in compliance with the Helsinki Declaration. Ethical approval was obtained from the Medical Ethical Council of Zhongshan Hospital, Fudan University (Approval number: B2021-020R). Informed consent was obtained from all the patients. Animal experiments were performed with the approval of the Institutional Animal Care and Use Committee at Fudan University (Approval number: 2021JS0076).

## Consent for publication
Not applicable.

## Competing interests
The authors declare that they have no competing interests.

## Author details
[1]State Key Laboratory of Common Mechanism Research for Major Diseases, Department of Biochemistry and Molecular Biology, Institute of Basic Medical Sciences Chinese Academy of Medical Sciences, School of Basic Medicine Peking, Union Medical College, Beijing 100005, China. [2]State Key Laboratory of Genetic Engineering, School of Life Sciences, Fudan University, Shanghai 200438, China. [3]Institute of Clinical Medicine, Peking Union Medical College and Chinese Academy of Medical Sciences, Translational Medicine Center, Peking Union Medical College Hospital, Beijing 100730, China. [4]State Key Laboratory of Pharmaceutical Biotechnology, School of Life Sciences, Chemistry and Biomedicine Innovative Center, Nanjing University, Nanjing 210023, China. [5]School of Basic Medical Sciences, Jiangxi Medical College, Nanchang University, Nanchang 330031, China. [6]Institute of Respiratory Disease, The First Affiliated Hospital of Nanchang University, Nanchang 330006, China. [7]Institute of Organoid Technology, Kunming Medical University, Kunming 650500, China.

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

## 