## [**Additional file 3. **Review history. · Genome Biology]

Review History

First round of review

Reviewer 1

Are you able to assess all statistics in the manuscript, including the appropriateness of statistical tests used? Yes, and I have assessed the statistics in my report.

Comments to author:

This is a nicely done study. The problem is significant, and the approach is well-performed. Elegant design of screening, robust sequencing and data analysis, and validation.

some comments to consider addressing:

1. the organic screen is unlike cell pool screen, heterogeneity is a challenge, how can the authors ensure the mosaic transduction does not interfere with screening?
2. the hits like FOS and Urb5 are interesting, but the follow up is a bit preliminary. the authors may miss a lot of opportunities by staying at this stage. why not follow up deeper on the mechanism?
3. statistics need to be better described in legends, methods, or provide detail excel tables

Reviewer 2

Are you able to assess all statistics in the manuscript, including the appropriateness of statistical tests used? No, I do not feel adequately qualified to assess the statistics.

Comments to author:

Liang and colleagues present a single-cell CRISPR screening assay focused on mouse intrahepatic cholangiocyte organoids, termed OSCAR, to screen for hepatocyte differentiation factors. In this process, the authors identify key roles for the Fos and Ubr5 genes. In follow-up experiments, Ubr5 is further characterized in in vivo experiments. While CRISPR screens have been reported before in organoids, this study presents a technological advance in the organoid field, as it establishes a screen not reliant solely on proliferation but instead screens for cell fate regulators. The manuscript is well written, the figures are clear to follow, and the results are exciting.

I do not have any technical concerns about the CRISPR screen. In general, a question that arises is whether the authors are truly screening for hepatocyte differentiation factors, or rather for cholangiocyte transdifferentiating factors towards hepatocyte-like features. Since the authors use ICOs they are screening for genes enabling an identity switch. If the authors would want to make claims that the novel genes are involved truly in hepatocyte differentiation, follow-up experiments would be recommended, e.g. in iPSC cells (which could also be of human origin). At the minimum, a discussion on this aspect is warranted.

Some more characterization of the Fos-KO organoids is needed. What are the basic features of these Fos KO organoids? How are the expression levels of the hepatocyte genes in expansion medium, in other words is it only active during "differentiation" or does it simply change cell fate? Do they grow at the same rate? What about expression of cholangiocyte markers versus the controls? The methodological approach of the authors is not ideal here, as it would be preferred to make a stable KO line, which can be characterized and then later on differentiated. If methodologically complex, in an alternative manner the authors could make use also of the FOS inhibitor to address these points.

The Ubr5 findings are interesting but remain rather elusive. The authors detect a set of amount of differentially expressed genes and highlight only a few. The authors could provide GO-term enrichment analysis on these genes to unbiasedly show the changes and provide a list of the genes. Regarding its putative function, the authors hypothesize that this could be due to hedgehog signaling. Did the authors identify any hints for this in their RNA-seq? Beyond gene function, are there any other notable changes on the liver? How do these mice and liver perform at later timepoints after birth?

Furthermore, what was the overlap between differentially expressed genes (amount and which genes) between the experiments on Ubr5 in the ICOs and in vivo?

Reviewer #1: This is a nicely done study. The problem is significant, and the approach is well-performed. Elegant design of screening, robust sequencing and data analysis, and validation.

some comments to consider addressing:

1. the organic screen is unlike cell pool screen, heterogeneity is a challenge, how can the authors ensure the mosaic transduction does not interfere with screening?

Response:

Thank you for your insightful comment. We agree that the heterogeneity of the organoid system poses a unique challenge for our screen. The heterogeneity arises from three aspects: Cells are present at different cell states in the organoids, each cell may have a different sgRNA transfected with mosaic transduction, and even the same sgRNA sequence could generate a heterogeneous population of cells, with its target gene unedited, or edited differently.

By leveraging single-cell sequencing data as the readout of the screen, we were able to identify heterogeneous states of cells in organoids as well as the heterogeneous states caused by mosaic transduction. To tackle the heterogeneity challenge, we have incorporated several steps into our analysis framework:

First, we used changes in the regulatory network as the functional readout for each cell, rather than specific gene expression changes. This allowed us to infer perturbation effects independent of the heterogeneous gene expression inherent in organoid cells.

Second, for the mosaic transduction, we determined sgRNA identity for each cell using direct detection of sgRNAs in the transcriptome data.

Thirdly, we also inferred the gene perturbation status from single-cell transcriptome data. Plotting the observed effects for each cell for all the cells with the same sgRNA, we observed that the cells followed a distinct pattern, indicative of a mix of edited and unedited cells. We excluded unedited cells for further analysis to ensure a precise assessment of perturbation effects.

We have added a paragraph in the discussion to clarify these steps for the heterogeneity challenge (Page 20):

“After identifying key regulators based on their overall regulatory profiles, we separated the perturbed and unperturbed cells based on their gene expression patterns, and removed the unperturbed cells for each sgRNA to ensure the accurate mapping of perturbation effects from a heterogeneous cell population in organoids. We were able to rank the regulators based on their direct impacts on the expression of hepatocyte marker genes.”

2. the hits like FOS and Urb5 are interesting, but the follow up is a bit preliminary. the authors may miss a lot of opportunities by staying at this stage. why not follow up deeper on the mechanism?

Response:

We appreciate your valuable suggestion. We are actively pursuing further experiments to understand the mechanistic roles of FOS and Ubr5 in hepatocyte formation and liver development. Preliminary findings have provided intriguing insights into their functions. In this revision, we have added experimental data regarding the defect in lipid metabolism in *Ubr5* conditional knockout mice (new Fig. 5g and Fig. S11). Our current manuscript primarily emphasizes the establishment of an in-organoid screening framework for cell fate regulators. Consequently, the detailed mechanistic data, while valuable, were considered beyond the immediate scope of this paper.

3. statistics need to be better described in legends, methods, or provide detail excel tables

Response:

We are grateful to the reviewer's feedback on the statistics description in our manuscript. We have added more detailed information about the statistical tests used in the legends and methods section. Furthermore, we have included an excel table (Additional file 2: Table S10) detailing the statistics associated with Fig. 4, Fig. 5, Fig. S1, Fig. S9 and Fig. S11.

In the methods section, we have added the following description of our statistical analysis (Page 33-34):

“Statistical analysis

The hypothesis tests in this paper were carried out using the 'stat' package function in R language (version 4.0.2) or Prism Software (version 6.0). For small sample dataset that did not pass the Shapiro-Wilk test (P -value < 0.05), logarithmic transformations were applied to enhance data normality. The student's t test was employed for comparisons between two groups, while one-way Analysis of Variance (ANOVA) were applied for comparisons involving more than two groups. Spearman's correlation analysis was used to quantify the correlation among datasets. The significance threshold of the P -value was set to 0.05 by default. Adjustments for multiple hypothesis testing were made using either the False Discovery Rate (FDR; achieved by Benjamini-Hochberg or q -value) or the Family-Wise Error Rate (FWER; achieved by Bonferroni), as specified in the manuscript.”

Reviewer #2: Liang and colleagues present a single-cell CRISPR screening assay focused on mouse intrahepatic cholangiocyte organoids, termed OSCAR, to screen for hepatocyte differentiation factors. In this process, the authors identify key roles for the *Fos* and *Ubr5* genes. In follow-up experiments, *Ubr5* is further characterized in in vivo experiments. While CRISPR screens have been reported before in organoids, this study presents a technological advance in the organoid field, as it establishes a screen not reliant solely on proliferation but instead screens for cell fate regulators. The manuscript is well written, the figures are clear to follow, and the results are exciting.

I do not have any technical concerns about the CRISPR screen. In general, a question that arises is whether the authors are truly screening for hepatocyte differentiation factors, or rather for cholangiocyte transdifferentiating factors towards hepatocyte-like features. Since the authors use ICOs they are screening for genes enabling an identity switch. If the authors would want to make claims that the novel genes are involved truly in hepatocyte differentiation, follow-up experiments would be

recommended, e.g. in iPSC cells (which could also be of human origin). At the minimum, a discussion on this aspect is warranted.

Response:

Thank you for your valuable feedback and insightful comments. We agree that the regulators we identified are factors enabling the identity switch from cholangiocyte to hepatocyte-like cells. While we have validated the roles for one of the factors, *Ubr5*, in *in vivo* liver development using a mouse model, we acknowledge that further experiments using iPSC cells are needed to confirm the involvement of these factors in human hepatocyte differentiation. We appreciate your suggestion and have included the point in the discussion (Page 22):

“Our screen was conducted using ICOs, which possess a cholangiocyte identity and could be converted to hepatocyte-like cells upon induction. The regulators we identified in the screen are factors that are involved in the cholangiocyte-hepatocyte identity switch. While we have validated the role of *Ubr5* in mouse liver development, additional studies employing the iPSC-to-hepatocyte differentiation system are crucial to ascertain these factors’ role in human hepatocyte differentiation and maturation.”

Some more characterization of the *Fos*-KO organoids is needed. What are the basic features of these *Fos* KO organoids? How are the expression levels of the hepatocyte genes in expansion medium, in other words is it only active during "differentiation" or does it simply change cell fate? Do they grow at the same rate? What about expression of cholangiocyte markers versus the controls? The methodological approach of the authors is not ideal here, as it would be preferred to make a stable KO line, which can be characterized and then later on differentiated. If methodologically complex, in an alternative manner the authors could make use also of the *FOS* inhibitor to address these points.

Response:

Thank you for your thoughtful comments. Following your suggestions, we have performed additional experiments to provide a detailed characterization of stable *Fos*-KO organoid lines.

Morphologically, the stable *Fos*-KO organoids were similar to *Fos*-intact control mICOs in both size and shape (see Additional file 1: Fig. S9a). Using the CellTiter-Glo Luminescent Cell Viability Assay, we found that the growth rates of *Fos*-KO organoids were also comparable to that of the control mICOs (see Additional file 1: Fig. S9b).

In the expansion medium (EM), we evaluated the expression levels of both cholangiocyte and hepatocyte markers in the *Fos*-KO organoids using qPCR. We found that the expression levels of cholangiocyte markers were high in the *Fos*-KO, with *Spp1*’s expression levels even higher than that of wild-type mICOs. The hepatocyte markers were detected at the low level comparable to non-targeting control mICOs. These results suggested that the knockout of *Fos* does not alter the cell fate for organoids cultured in the expansion medium.

Upon induction with differentiation media (DM), both non-targeting control (sgNT) and *Fos*-KO organoids displayed a significant reduction in cholangiocyte marker (*Sox9* and *Spp1*) expression, while the five hepatocyte markers (*Alb*, *Ttr*, *Apoa1*, *Mup20* and *Mrp2*) were significantly increased. Among them, four hepatocyte markers were expressed at significantly higher levels and both

cholangiocyte markers were markedly reduced in the *Fos*-KO organoids compared to their counterparts. These results suggested that while knockout of *Fos* does not change the cholangiocyte identity for organoids in expansion medium, it does enhance hepatocyte differentiation upon induction.

We have added the new results in Additional file 1: Fig. S9a-c and described them in the manuscript as follows (Page 14):

“Morphologically, *Fos* KO organoids closely resembled the NT mCOs (Supplementary Figure Additional file 1: Fig. S9a). Using the cell viability assay, we found that their growth rates were comparable (Additional file 1: Fig. S9b). In *Fos*-KO organoids cultured with expansion medium, cholangiocyte markers (*Sox9* and *Spp1*) were upregulated, while hepatocyte markers were barely detectable by qRT-PCR (Additional file 1: Fig. S9c). Upon induction with differentiation medium, globally, cholangiocyte markers were significantly reduced in NT and *Fos* KO organoids, while all hepatocyte markers (*Alb*, *Ttr*, *Apoa1*, *Mup20* and *Mrp2*) were significantly increased (Additional file 1: Fig. S9c). Interestingly, four of the five hepatocyte markers were expressed at significantly higher levels in the *Fos*-KO organoids. These results suggested that *Fos* is a conditional liver cell fate regulator, and the deletion of *Fos* significantly enhanced hepatocyte differentiation upon differentiation induction.”

The *Ubr5* findings are interesting but remain rather elusive. The authors detect a set of amount of differentially expressed genes and highlight only a few. The authors could provide GO-term enrichment analysis on these genes to unbiasedly show the changes and provide a list of the genes. Regarding its putative function, the authors hypothesize that this could be due to hedgehog signaling. Did the authors identify any hints for this in their RNA-seq?

Response:

Thank you for your valuable comments on the *Ubr5* findings. We have performed a GO-term enrichment analysis on the differentially expressed genes in *Ubr5* knockout liver. This analysis revealed significant enrichment in various metabolic pathways, especially those related to lipid metabolism, such as the olefinic compound metabolic process, the steroid metabolic process, and the long-chain fatty acid metabolic process (see new Fig. 5g).

Although we previously hypothesized that the association of hedgehog signaling with the role of *Ubr5*, our RNA-seq data did not indicate significant changes in genes directly linked to the hedgehog signaling.

Given the enrichment of lipid metabolic pathways among the differentially expressed genes, we used the Oil red O staining assay to evaluate the lipid accumulation in liver. We found that the red oil staining level were significantly increased in *Ubr5* knockout livers at 3 weeks compared to wild-type liver (see Additional file 1: Fig. S11c, d).

Based on these results and following your suggestions, we now included the GO enrichment analysis in Fig. 5g and Additional file 2: Table S8b, and the Oil red O staining results in Fig. S11c, d. We describe the results in the text as follows:

Page 17-18: “GO-term enrichment analysis on the 266 differentially expressed genes in *Ubr5* knockout liver revealed significant enrichment in metabolic pathways related to lipid metabolism, such as olefinic compound metabolic process, steroid metabolic process, and long-chain fatty acid metabolic process (Fig. 5f, g and Additional file 2: Table S8a, b). Indeed, indicated by Oil Red O staining, *Ubr5* knockout liver had increased lipid accumulation at 3 weeks compared to wild-type liver (Additional file 1: Fig. S11c, d), suggesting a specific role of *Ubr5* in regulating the lipid metabolic pathways.”

Page 21: “Our findings that *Ubr5* ablation delayed hepatocyte differentiation and maturation, revealed the significance of this regulator in liver development. The transcriptional profiling revealed a strong defect of lipid metabolism, which was further supported by the increased lipid accumulation observed in *Ubr5* knockout liver at 3 weeks. The potential molecular mechanism underlying *Ubr5*'s regulation on hepatocyte differentiation needs further investigations. Given that *Ubr5* is an E3 ligase, the finding of *Ubr5*'s substrates and its interactions with other proteins might provide more insights. “

Beyond gene function, are there any other notable changes on the liver? How do these mice and liver perform at later timepoints after birth?

Response:

Although we have observed defects in lipid metabolism, the *Ubr5* conditional knockout (CKO) mice were viable after birth and exhibited no significant abnormalities up to 21 days of age. The size of the liver in *Ubr5* CKO mice was also comparable to that of control mice, with no notable changes in liver morphology observed during this period.

To provide a more comprehensive view, we have included images of the livers from *Ubr5* CKO mice in Fig. S11a and have elaborated on this in the text (Page 17):

“The *Ubr5* knockout liver's size was comparable to *Ubr5* WT, with no notable changes in liver morphology or structure observed up to 21 days”.

We are in the process of performing follow-up studies to monitor the mice and their liver functions at later timepoints after birth. Preliminary results suggest that these mice could survive beyond 60 days. We will detail these findings in our future publications.

Furthermore, what was the overlap between differentially expressed genes (amount and which genes) between the experiments on *Ubr5* in the ICOs and in vivo?

Response:

Thank you for your valuable suggestion. We have compared the differentially expressed genes in *Ubr5* KO ICOs and those in *Ubr5* knockout liver. We identified an overlap of 27 genes that were upregulated in both *Ubr5* knockout models. These genes include *Igf2*, *Lrtm2*, *Gm36283*, *H1f10*, *Meg3*, *Afp*, *Ces2c*, *Bicc1*, *Ldhd*, *Grip1*, *Cyp2c55*, *H2bc6*, *Slc4a3*, *Ces1f*, *Adcy1*, *Gpc3*, *St3ga16*, *H1f4*, *Nrg1*, *Spp1*, *Robo1*, *Auts2*, *Acot2*, *Fgr*, *Syk*, *Pltp*, and *Dio3os*. Conversely, 7 genes were down-regulated in both models: *Grh1*, *Mup20*, *Sult5a1*, *Prom2*, *Hsd17b2*, *Gadd45g*, and *Seprina9*.

The overlap differentially expressed genes are consistent with the role of *Ubr5* in hepatocyte cell fate determination. Notably, the premature marker of hepatocyte *Afp* and the cholangiocyte marker *Spp1*

was upregulated in both knockout models. Meanwhile, key liver-specific metabolic enzymes, such as *Mup20* and *Sult5a1* were significantly reduced in both knockout models.

We have incorporated the overlap results in the manuscript (Page 18):

"Among the 39 genes that were downregulated in *Ubr5* knockout, 7 genes were also significantly down-regulated in *Ubr5* knockout mICOs compared to their wild-type counterpart, including several classical hepatocyte markers (*Sult5a1* and *Mup* members) (Fig. 5f and Additional file 2: Table S8c)."

We hope that these revisions address the reviewer's concerns and improve the overall quality of our paper. Thank you for your valuable feedback.

Second round of review

Reviewer 1

Previous comments well addressed.

Reviewer 2

The authors have done a great job at addressing my comments. I have no further comments. Congratulations to the authors for this exciting manuscript.